environmental science, ecology

long-term species decline, land-use change, burrowing mammal, agricultural conversion, Corona spy satellite imagery

**Author for correspondence:**
Catalina Munteanu
e-mail: catalina.munteanu@geo.hu-berlin.de

# Cold War spy satellite images reveal long-term declines of a philopatric keystone species in response to cropland expansion

Catalina Munteanu[1,4,5], Johannes Kamp[6], Mihai Daniel Nita[5], Nadja Klein[2], Benjamin M. Kraemer[7], Daniel Müller[1,3,4], Alyona Koshkina[6,8], Alexander V. Prishchepov[9,10] and Tobias Kuemmerle[1,3]

[1]Geography Department, [2]Department of Statistics, and [3]Integrative Research Institute for Transformations in Human-Environment Systems, Humboldt University Berlin, Unter den Linden 6, 10099 Berlin, Germany
[4]Leibniz Institute of Agricultural Development in Transition Economies (IAMO), Theodor Lieser Straße 2, 06120 Halle (Saale), Germany
[5]Department of Forest Engineering, Faculty of Silviculture and Forest Engineering, Transylvania University of Brasov, 1 Sirul Beethoven, Brasov, Romania
[6]Institute of Landscape Ecology, University of Münster, Heisenbergstrasse 2, 48149 Münster, Germany
[7]Leibniz Institute of Freshwater Ecology and Inland Fisheries, Müggelseedamm 310, 12587 Berlin, Germany
[8]Association for the Conservation of Biodiversity of Kazakhstan (ACBK), 18 Beibitshilik Street, Office 406, Astana 010000, Kazakhstan
[9]Department of Geosciences and Natural Resource Management (IGN), University of Copenhagen, Øster Voldgade 10, 1350 København K, Denmark
[10]Institute of Steppe of the Ural Branch of the Russian Academy of Sciences, Pionerskaya Street 11, Orenburg 460000, Russia

CM, 0000-0003-1616-9639; JK, 0000-0002-8313-6979; MDN, 0000-0002-6072-7784; BMK, 0000-0002-3390-9005; DM, 0000-0001-8988-0718; AK, 0000-0002-2501-1887; AVP, 0000-0003-2375-1651; TK, 0000-0002-9775-142X

Agricultural expansion drives biodiversity loss globally, but impact assessments are biased towards recent time periods. This can lead to a gross underestimation of species declines in response to habitat loss, especially when species declines are gradual and occur over long time periods. Using Cold War spy satellite images (Corona), we show that a grassland keystone species, the bobak marmot (*Marmota bobak*), continues to respond to agricultural expansion that happened more than 50 years ago. Although burrow densities of the bobak marmot today are highest in croplands, densities declined most strongly in areas that were persistently used as croplands since the 1960s. This response to historical agricultural conversion spans roughly eight marmot generations and suggests the longest recorded response of a mammal species to agricultural expansion. We also found evidence for remarkable philopatry: nearly half of all burrows retained their exact location since the 1960s, and this was most pronounced in grasslands. Our results stress the need for farsighted decisions, because contemporary land management will affect biodiversity decades into the future. Finally, our work pioneers the use of Corona historical Cold War spy satellite imagery for ecology. This vastly underused global remote sensing resource provides a unique opportunity to expand the time horizon of broad-scale ecological studies.

## 1. Background

Agriculture is essential for human societies, but millennia of agricultural land-use changes have transformed much of the planet's land surface and contributed to the ongoing biodiversity crisis [1–3]. The world's grasslands are particularly affected by agriculture, with as much as 80% lost on some continents [4]. This loss is worrisome because grasslands harbour astonishing biodiversity of plants, insects, birds, and large grazers—American bison and pronghorn antelope in North America;

Saiga antelope, Asiatic wild ass, and Mongolian gazelle in Eurasia; and wildebeest and zebras in Africa [5–7]. Understanding the biodiversity response to agricultural expansion relies largely on satellite remote sensing [8]. But remote sensing assessments are restricted to recent time periods, potentially missing responses to past land conversions.

Recent advances in remote sensing have improved the understanding of grassland dynamics over the past three decades, including agricultural expansion and intensification, abandonment, and grassland degradation [9–11]. But widespread conversion of grasslands occurred long before the emergence of now established remote sensing approaches [3,12]. During European settlement in North America, large tracts of the Great Plains were converted to agriculture, causing land degradation that peaked in the Dust Bowl of the 1930s [13,14]. Across the Eurasian steppes, following the World War II food crisis, huge swaths of grasslands were converted to croplands [14]. Yet, spatially explicit, fine-scale land-use data are rarely available for large areas from before the 1980s. This limits the recognition of biodiversity responses to mid-twentieth century land-use changes.

Land conversions can lead to gradual or time-delayed declines in diversity and abundance because species may require some time following disturbances, until they reach a new equilibrium [15,16]. Land conversion can create population sinks, where local extinctions occur within years [17–19], decades, or centuries [19,20]. The speed and timing of population declines may depend on the spatial configuration of remaining habitat and life-history traits, such as longevity [21]. Furthermore, agricultural practices may affect population fitness and lower forage availability leading to lower recruitment or survival over time [22–24]. This is why long-term population assessments following historical land conversions are essential to understand the full effects of conversions on species.

Burrowing rodents, such as marmots, ground squirrels, maras, and wombats, are critical for assessing long-term ecosystem functioning because they are keystone species and ecosystem engineers [25]. Burrowing rodents provide dens, nesting habitat, and shelter for many other species, such as foxes, owls, and arthropods [26,27]. Rodents are a food source for larger predators, and through digging and herbivory, they increase soil nitrogen content and forage quality for large grazers [25]. However, human activities have caused major declines in burrowing rodent populations worldwide, directly through poisoning or hunting and indirectly through agricultural expansion and intensification [23,28]. The repeated disturbance of the burrows through agricultural practices (i.e. tillage, harvest, pesticide application) might lead to population fitness declines, ultimately causing a population drop [22,23]. Many burrowing rodents exhibit philopatric behaviour [29], meaning that their dispersal is constrained either by life history or ecological factors. Philopatry makes burrowing rodents an interesting focus of long-term land change studies, because it may constrain their responses to environmental change, creating ecological traps [30]. Changes in rodent abundance and community composition may have large cascading effects on grassland ecosystems. Yet, despite their disproportional importance, long-term population dynamics of most burrowing rodents are understudied [23,25].

High-resolution satellite images represent a reliable and cost-effective resource for detecting burrowing animal occurrence, distribution, and abundance [31], including wombats in Australia, prairie dogs in the USA, and marmots in Central Asia [32,33]. However, high-resolution satellite imagery is typically only available since the 2000s, precluding long-term studies. A remarkable but largely untapped resource to enable long-term studies are Cold War spy satellite imagery from the Corona missions [34,35], which provide high-resolution imagery back to the 1960s with global coverage [36]. This imagery is valuable for many applications in ecology and conservation, particularly when integrated with contemporary data. Yet, no study has made use of the tremendous potential of Corona for long-term biodiversity studies.

The Eurasian steppe experienced drastic episodes of land-use change in the twentieth century during the rise and fall of the Soviet Union. The steppes of Kazakhstan, at 36% of their historical extent, are the largest remaining continuous area of Eurasian steppe [37]. Much of the Kazakh steppe was converted to cropland in the mid-twentieth century during the Virgin Lands Campaign (1954–1963) [38,39], as a result of Soviet policies to increase domestic food production following World War II. Much of this cropland was abandoned after the Soviet Union collapse. Finally, substantial recultivation of abandoned fields and cropland intensification occurred after 2005 [10,40], due to policy reforms and rising global cereal prices [38]. Together, this provides a unique natural experiment for understanding the ecological responses of burrowing mammals to agricultural conversions.

Our goal was to understand how bobak marmots, a keystone species of Eurasian steppes, responded to agricultural conversions since the 1960s. Specifically, we asked: (i) *How did the marmot population respond to cropland expansion since the 1960s?* and (ii) *how did the species choice of burrow location change with changing land use?* We expected that burrow density would be higher in grasslands than croplands, because grasslands represent the species' natural habitat. We also expected marmot declines where cropland expanded, but not in persistent grasslands where burrows were not ploughed. Finally, we expected that agricultural expansion caused a spatial redistribution of marmot burrow locations.

## 2. Material and methods

To address our two questions, we mapped over 12 500 marmot burrows, and related them to the surrounding land use and to other environmental and anthropogenic factors for two points in time, a *historical* period (1968–1969, based on Corona imagery) and a *contemporary* period (1999–2017, based on Google Earth, Bing, and Esri imagery) for a random sample of 900 plots of 1 km diameter, together covering an area of 60 000 km$^2$ (figure 1).

### (a) Study area

We studied marmot burrow occupancy and density in northern Kazakhstan along a north–south gradient, mirroring natural latitudinal gradients of climate, soil, and land use, in the provinces of northern Kazakhstan (site 1), Kostanay and Aqmola (site 2), and Qaragandy (site 3), all within the contemporary and historical ranges of steppe marmots in Kazakhstan (figure 1). The natural vegetation of northern Kazakhstan are forest steppes, steppes, and semi-deserts [14], but today the area is largely used for rainfed agriculture, primarily to grow wheat (hereafter cropland) [38] and livestock grazing (hereafter grassland). After the collapse of the Soviet Union in 1991, livestock numbers collapsed and grazing on large pasture areas stopped [41]. Our study region had approximately 44% grassland cover in the historical time period and 38% in the contemporary period (figure 2). Cropland extent

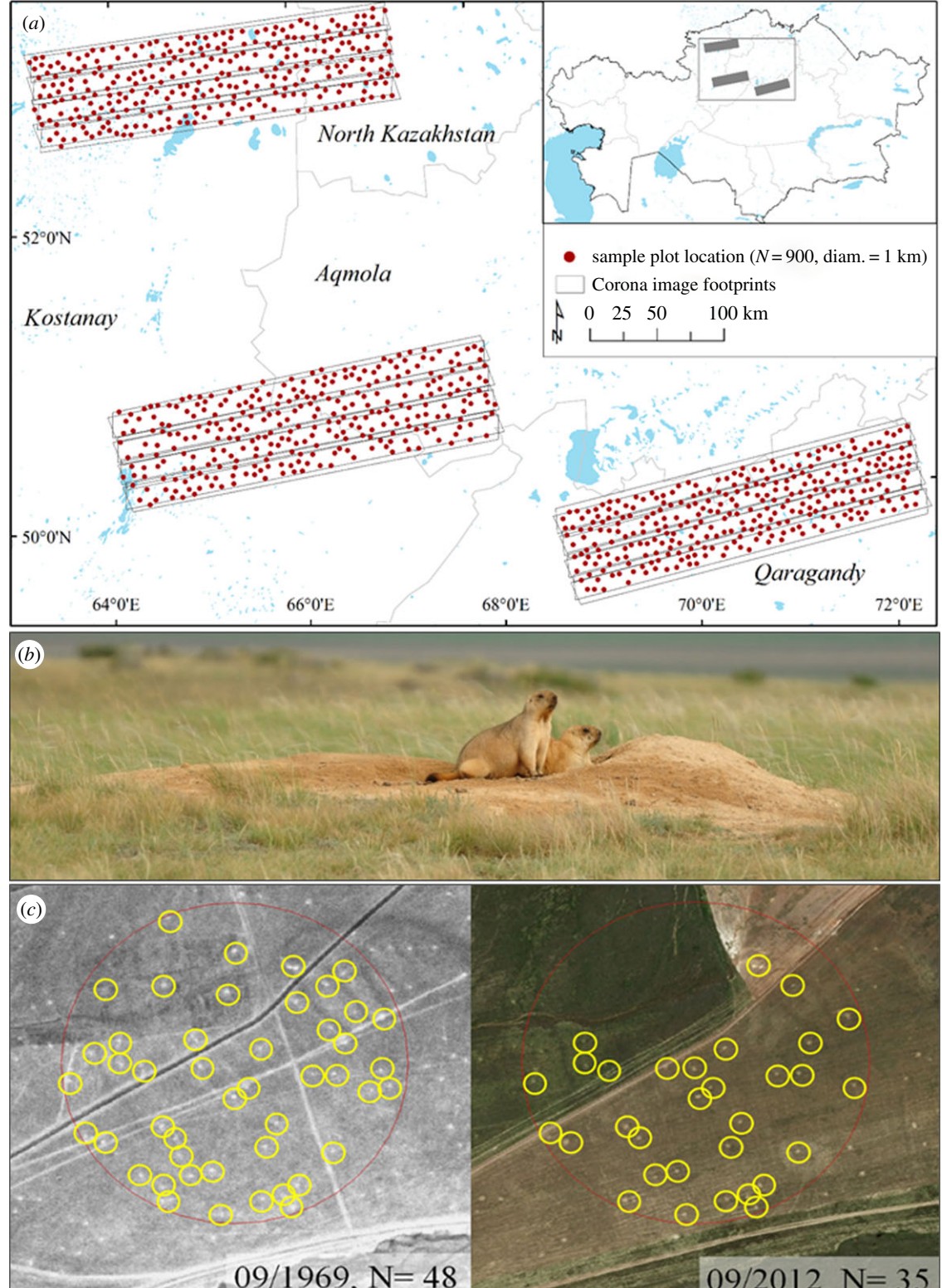

**Figure 1.** Study area in northern Kazakhstan, covered by 12 Corona image footprints (*a*). Marmot burrows can be detected from space because they are round, large areas of bare soil (Photo credit: A.K., 2015) (*b*). For each sample plot (1 km diameter), we derived the number and location of all burrows for a historic and a contemporary time period (*c*). (Online version in colour.)

increased between the two time periods but as much as 34% of the contemporary cropland was fallow or abandoned.

Following World War II food shortages across the Soviet Union, the Virgin Lands Campaign (1954–1963) led to the conversion of 25.5 million ha of steppe to cropland (mostly wheat) in Russia and Kazakhstan [38]. Despite being a short-term success, this campaign affected the steppe ecosystem functioning in major ways through deep ploughing of steppe soils, salinization, and increased wind erosion [39,42]. In addition, habitat for many species was reduced [43]. After

the Virgin Lands Campaign, agricultural expansion slowed, and croplands reached maximum extent in the late 1980s. When state support contracted after the fall of the Soviet Union, frequent droughts led to the abandonment of as much as 50% of croplands in Kazakhstan in the 1990s [10]. Many areas have been re-cultivated in recent decades following increasing world market prices for cereals, improved institutional conditions, and technological progress, such as the adoption of no-till agriculture [10,40]. The Post-Soviet abandonment trend may provide new opportunities for steppe conservation, but

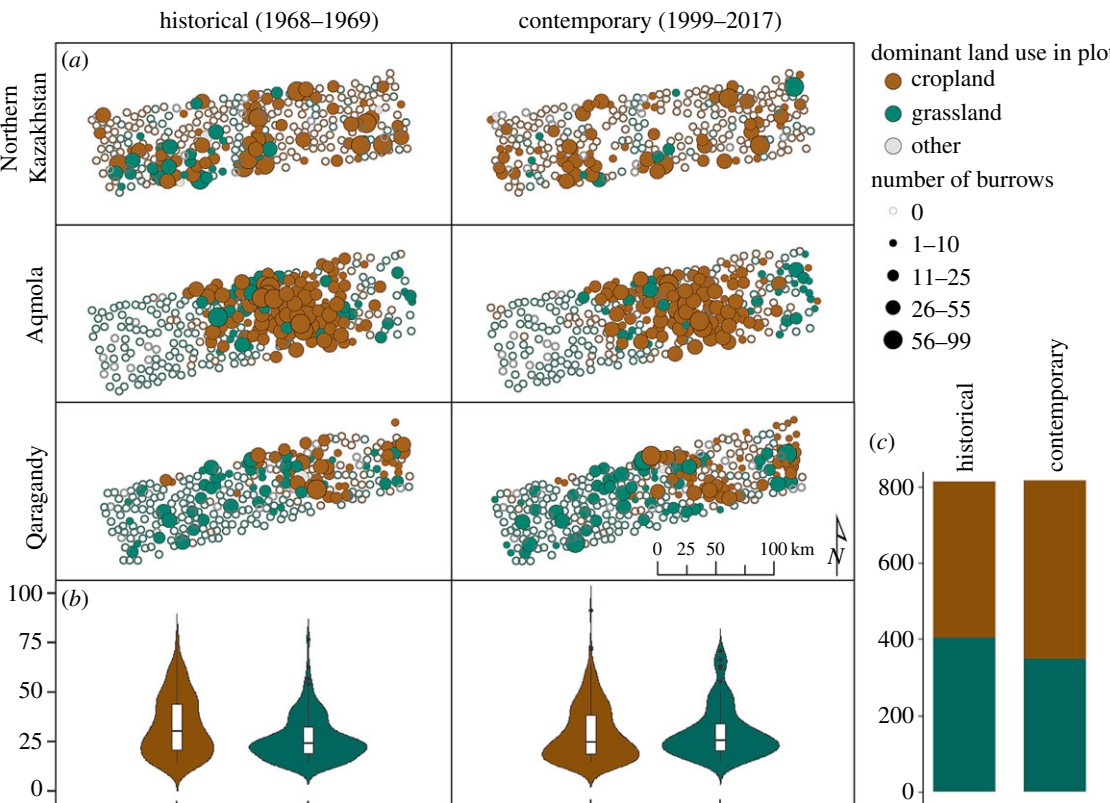

**Figure 2.** Marmot burrow distribution across space, time, and land-use classes. Land use and burrow density during the historical (1968–1969) and contemporary (1999–2017) time periods (*a*). Observed number of burrows per plot for cropland and grassland plots (*b*). Land use in plots for which we could distinguish cropland or grassland use (*N* = 814, 50% cropland historically, 57% cropland contemporary) (*c*). (Online version in colour.)

these are diminishing as agricultural recultivation and transition to no-till agriculture have increased since the early 2000s [38,41].

## (b) Corona imagery and land-use data

We derived land-use information, marmot occupancy, and densities from two main satellite image sources, each corresponding to a major episode of land use in the region: (i) historic US spy satellite data from 1968 to 1969 to capture the conditions following the Virgin Lands Campaign (hereafter: *historical data*) and (ii) high-resolution, satellite imagery from the GoogleEarth™, ESRI Satellite™, and Bing™ platforms, dated 1999–2017, to capture the conditions after Soviet Union collapse (hereafter: *contemporary data*; figure 2).

For the historical time-layer, we obtained 12 pairs of stereographic filmstrips from the Corona missions (via earthexplorer.usgs.gov, see electronic supplementary material, S1). These images were taken in September 1968 and 1969, and are available as panchromatic, stereographic image strips, each covering about 17 × 230 km on the ground at resolutions ranging between 2.29 and 2.41 m [36]. We rectified the Corona images using structure-from-motion algorithms implemented in AgiSoft Photoscan™ [34]. The accuracy of the rectification was high, with average positional errors of 9.78 m (range: −47 m, +48 m) on the *x*-axis and 16.61 m (range: −56 m, +78 m) on the *y*-axis. For the contemporary imagery, we used three online map engines: ESRI Satellite Base Map™ (69% of the data), GoogleEarth™ (23% of the data), and Bing™ (8% of the data). Imagery was acquired between 1999 and 2017, but the majority of the data (70%) were dated between 2012 and 2014. Image resolution varied between 0.6 and 5 m.

To map historical and contemporary land use, we used a random sampling design of 900 plots (300 per area) within the marmots range [23], with a minimum distance of 5 km, to avoid spatial autocorrelation. Each sample plot had a 1 km diameter (plot area = 78.53 ha). For each plot and time period, we recorded the dominant land use, focusing on cropland and grassland, which

are the main habitats in which marmots occur in Kazakhstan. All other classes (settlement, forest, water, bare) were summarized into a single 'other' class. We performed automated image segmentation using eCognition software [44], and assigned corresponding land-use classes manually to each segment.

## (c) Marmot burrows and their spatial distribution

For each plot and each period (historical and contemporary), we recorded the number of detected marmot burrows (as proxies for marmot occupancy and density) at a working scale of 1 : 5000 to ensure the consistency of observed elements between individual plots (figure 1). We dropped 37 historical and 43 contemporary plots for which we could not detect land use or burrows due to cloud cover or high image distortion (less than 5% of all plots). Burrows appear as bright spots in both historical and contemporary imagery due to the large amount of soil turned by the marmots when digging and tending to the burrow (figure 1). Burrow location validation with field visits suggested that no false negatives occurred [33]. False positives only occurred in recently abandoned colonies (where burrows are usually covered by darker vegetation than the surrounding areas), but these were extremely scarce in our study area [33]. Overall, only *ca* 40% of the burrows on the ground are detectable with remote sensing, likely because temporary summer burrows are small [33]. In total, 36% of our samples (622 plots) had burrows. We hand-digitized a total of 12 607 burrows (of which 52% were from Corona imagery). Plots where burrows were present had an average density of 18.2 burrows/plot (19.4 for the historical periods, 17.1 for the contemporary period). Burrows occurred more often on cropland (47% of the cropland plots) than on grassland (26% of the grassland plots) (figure 2).

To ensure temporal consistency in the spatial distribution of burrows within a plot, we used a back-dating approach, commonly employed in digitizing information from historical maps, in which the location of the digitized element is verified

in subsequent time periods [45,46] in relation to stable landscape elements or, in our case, the spatial pattern of neighbouring burrows. We considered a marmot burrow to indicate philopatry if it was found at the same location in both the historical and the contemporary time period. For each plot, we assessed the number of burrows lost, persistent and new in relation to the number and location of burrows in the historical time period.

## (d) Modelling marmot burrow occurrence, density, and spatial distribution

We modelled marmot burrow density to evaluate the effect of land-use change on marmot population. Our sample size consisted of 1720 sample plots, of which 863 were entries for contemporary data and 857 for historical data (figure 2). We used one model to explain burrow occurrence and density (response variable = number of burrows per plot) across land uses and time and three additional models based on paired observations for the two time periods. The three additional models explained the occurrence and density of burrows lost (response variable = number of burrows lost per plot), new (response variable = number of burrows gained per plot), and the number of persistent burrows relative to the initial number of burrows a plot had in the historical time period. Persistent burrows refer to those burrows that were found at the exact same location in both time periods (response variable = number of persistent burrows per plot).

Aside from historical and contemporary land use and the initial number of burrows (for the latter three models), we considered variables that are known to influence marmot occupancy and density, such as soil texture, climate, vegetation, proximity to water, human activity, and terrain slope [23,26] (electronic supplementary material, S2). Soil texture accounted for marmot preference towards soils that are easy to dig in, but stable enough to maintain burrow structure. We used normalized difference vegetation index (NDVI) measures for the month of May, shortly after marmots emerged from hibernation, as a proxy for food availability [33]. For each plot, we calculated Euclidean distances to the nearest river because marmots burrow along higher river banks and avoid areas with near-surface ground water [23]. The distance to the nearest farm or livestock concentration point accounted for potential grazing interactions with livestock [33]. The average plot slope was accounted for because marmots prefer flat areas, with wide views that are advantageous for detecting predators. Last, we considered three climatic variables that are known to affect the species: summer temperature, summer precipitation, and winter temperature to account for marmot survival during hibernation (electronic supplementary material, S2). Because summer temperatures and winter temperatures were correlated (Pearson correlation $r = -0.59$), we only retained the second variable in our models. All other variables were only weakly correlated ($r < 0.55$; electronic supplementary material, S3). Time-variant predictors included land use and distance to farms (electronic supplementary material, S2). Climate data were fed into the model as mean values since the 1960s [33] because climate change was negligible in Kazakhstan for this time period, particularly regarding climate effects on crop yields [47]. We standardized and centred all variables to improve the model interpretability.

We fitted generalized linear mixed models with a zero-inflated negative binomial (ZINB) distribution using the 'glmmTMB' package in R [48], following assessment of several modelling approaches (electronic supplementary material, S4). The ZINB model best allowed us to quantify differences between land-use classes while accounting for zero-inflation and overdispersion. ZINB models also allow for two possible sources of zeros in the data: general environmental unsuitability for marmot presence (binomial part of the model) and (pseudo-)absences induced by processes affecting occurrence at the local scale (count part of the model). We expected that burrow occurrence is governed by

suitability factors such as climate, topography, or food resources, while density is largely driven by local and species-specific factors such as proximity to grazing livestock or food availability. Because we were specifically interested in quantifying the effect of time and land use for both presence and abundance, we included these two variables and their interaction in both the presence–absence and the abundance part of the model. As the plots were surveyed in the historical and the contemporary periods, we corrected for repeated measures by fitting sample plot id as a random effect, nested within study area. Random effects were used to account for unobserved heterogeneity not contained in the covariates.

To estimate the probability of burrow occurrence and the burrow density per plot, we used a total of 1720 observations (plots) from both time periods (electronic supplementary material, S2 and S5). We estimated the effects of land use and time on the probability of occurrence and on burrow density, while keeping all other variables at their mean values (electronic supplementary material, S6). To estimate the proportion of burrows that were lost, persistent, and newly created within a plot, we used a total of 843 observations paired by plot and time period (figure 1). For each plot, we considered the initial number of burrows in the historical time period and the major land-use changes that occurred between the two periods (persistent cropland, grassland to cropland, and persistent grassland), in addition to environmental and anthropogenic covariates. For each of the three models (lost, persistent, newly created), we estimated the effects of land change on the probability of occurrence and on burrow density for a plot that started out with 19 burrows (mean value for historical period), while keeping all other variables at their mean values (electronic supplementary material, S7).

## (e) Robustness check and comparison with further datasets

For a subset of cropland plots, we separated active and abandoned cropland ($N = 165$) and compared burrow density change in relation to transitions between these classes. For these plots, we observed no significant difference in the change in burrow numbers between the two classes. Because the separation between fallow and active agriculture is often not possible based on visual image interpretation, and because we observed no significant difference in our subsample, we combined active cropland and abandoned/fallow cropland into a general cropland class for all subsequent analyses (electronic supplementary material, S8).

To identify if the burrow density decline over the 50 years was gradual or abrupt, we carried out two analyses on data subsets. First, for 111 plots where archival map data were available, we compared the average number of burrows in plots where agriculture expanded during the Virgin Lands Campaign (1954–1963) with areas where agriculture was already established prior to the campaign (electronic supplementary material, S9). For these 111 plots, we compared the average number of burrows in the historical and contemporary time periods among two groups: plots that were converted to agriculture during the Virgin Lands Campaign and plots that were cropland already prior to the campaign. Second, for a subset of 138 plots which were classified as cropland in the contemporary time period, we obtained additional multi-temporal imagery between 2000 and 2019. We found no substantial changes in burrow density between 2000 and 2019, which discounts the possibility of a recent abrupt decline (electronic supplementary material, S10).

# 3. Results

## (a) Steepest declines in the oldest croplands

Marmot burrow densities were higher in croplands than in grasslands, but strong declines occurred over the past 50

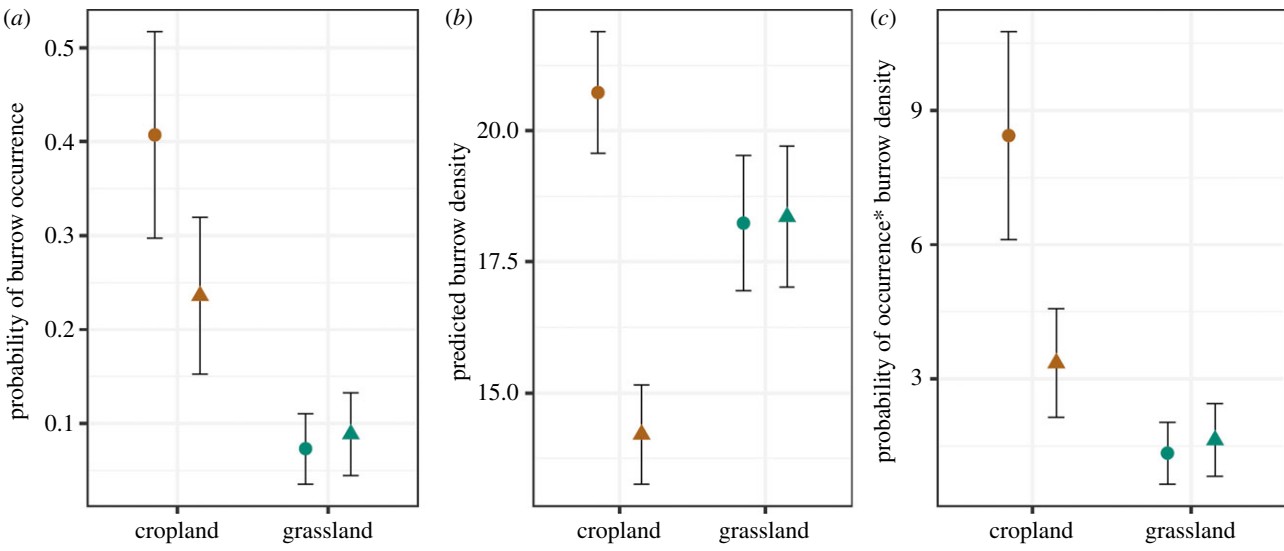

**Figure 3.** Predicted probability of burrow occurrence (zero-inflated model component) for croplands and grasslands, when keeping all other variables at their mean value. (*a*) Predicted number of burrows per plot, given burrows were present (count component of the model). (*b*) Expected mean number of burrows per plot, for croplands and grasslands. (*c*) Circles represent predictions for the historical time period and triangles predictions for the contemporary period. Error bars represent ±1 s.e. (Online version in colour.)

years (14% of the observed initial number of burrows). Most importantly, these declines were most prominent in persistent croplands and in those plots where cropland persisted the longest. Overall, our dataset indicated that burrow numbers decreased by 14% ($N = 1027$) since the 1960s (range: −60 to 55 burrows/plot) and we recorded burrow density decreases in 55% of the plots (figure 2). Surprisingly, our models revealed that the probability of occurrence was higher in croplands compared to grasslands, independent of time period (figure 3*a*). After accounting for zero-inflation, overdispersion, and environmental and human factors that may affect the burrow site selection by the marmots, we estimated higher burrow density in croplands compared to grasslands on average (figure 3*c*; electronic supplementary material, S11). However, most of the decline occurred in croplands, where the expected number of burrows dropped from 8.43 (±2.3) burrows compared to 3.35 (±1.2) burrows since the historical time period (figure 3*c*; electronic supplementary material, S11). Our model predicted a very small gain in grasslands (on average, less than 1 additional burrow per plot; figure 3*c*; electronic supplementary material, S6 and S11). This suggests that approximately 60% of the historical burrows were lost in croplands, whereas grasslands that persisted since the 1960s gained about 17% of the historical burrows.

Using ancillary information on cropland use prior to the Virgin Lands Campaign, we estimated that 17% of the agricultural fields identified in Corona images had already been converted to cropland prior to the Virgin Lands Campaign. Declines in plots that had been cropland since the early twentieth century were steeper ($N = 16$, declines of 78%) than in plots that were converted to cropland only during or after the Virgin Lands Campaign ($N = 95$, declines of 16%) (electronic supplementary material, S9).

### (b) Remarkable long-term persistence of marmot burrows

Our analyses revealed remarkable long-term persistence of marmot burrows despite drastic land-use change, suggesting

a high degree of site-conservatism and philopatry in steppe marmots. Despite land-use change, the majority of plots we assessed had at least some burrows at exactly the same locations as in the historic period (i.e. persistent burrows figure 4*a*). In other words, approximately eight marmot generations maintained the same burrows. Of all historical burrows, at least some burrows were recorded at the same locations in 62% of all studied plots (figure 4*a*). Our models predicted that persistent cropland plots (i.e. plots that were converted to cropland during or prior to the Virgin Lands Campaign) lost a higher proportion of burrows (62% ± 6%) compared to stable grasslands plots (40% ± 5%), and had a lower proportion of maintained burrows (figure 4*d*). Specifically, for a hypothetical plot that had 19 burrows initially, we estimated that in persistent grasslands, approximately 33% of the historical burrows were maintained, suggesting philopatry of their denizens compared to only 22% in croplands (figure 4*d*; electronic supplementary material, S7 and S11). This relationship was consistent, regardless of the initial burrow number, but the differences were even larger for plots that had higher numbers of initial burrows (electronic supplementary material, S7).

### 4. Discussion

Impact assessments of agricultural expansion on biodiversity typically focus on the time immediately following habitat loss, which is problematic if biodiversity changes are gradual over long time periods. We reveal one of the longest recorded responses of a mammal to historical agricultural conversion and highlight that single snapshots in time may provide insufficient information for understanding how species respond to land conversions. Our analysis of changes in marmot burrow densities since the 1960s suggests that bobak marmot populations declined as a result of past habitat conversion, and that these declines occurred on timescales of up to 50 years. Burrow declines were steepest in persistent cropland, indicating that declines are a long-term, gradual response to historical agricultural conversions [38], related

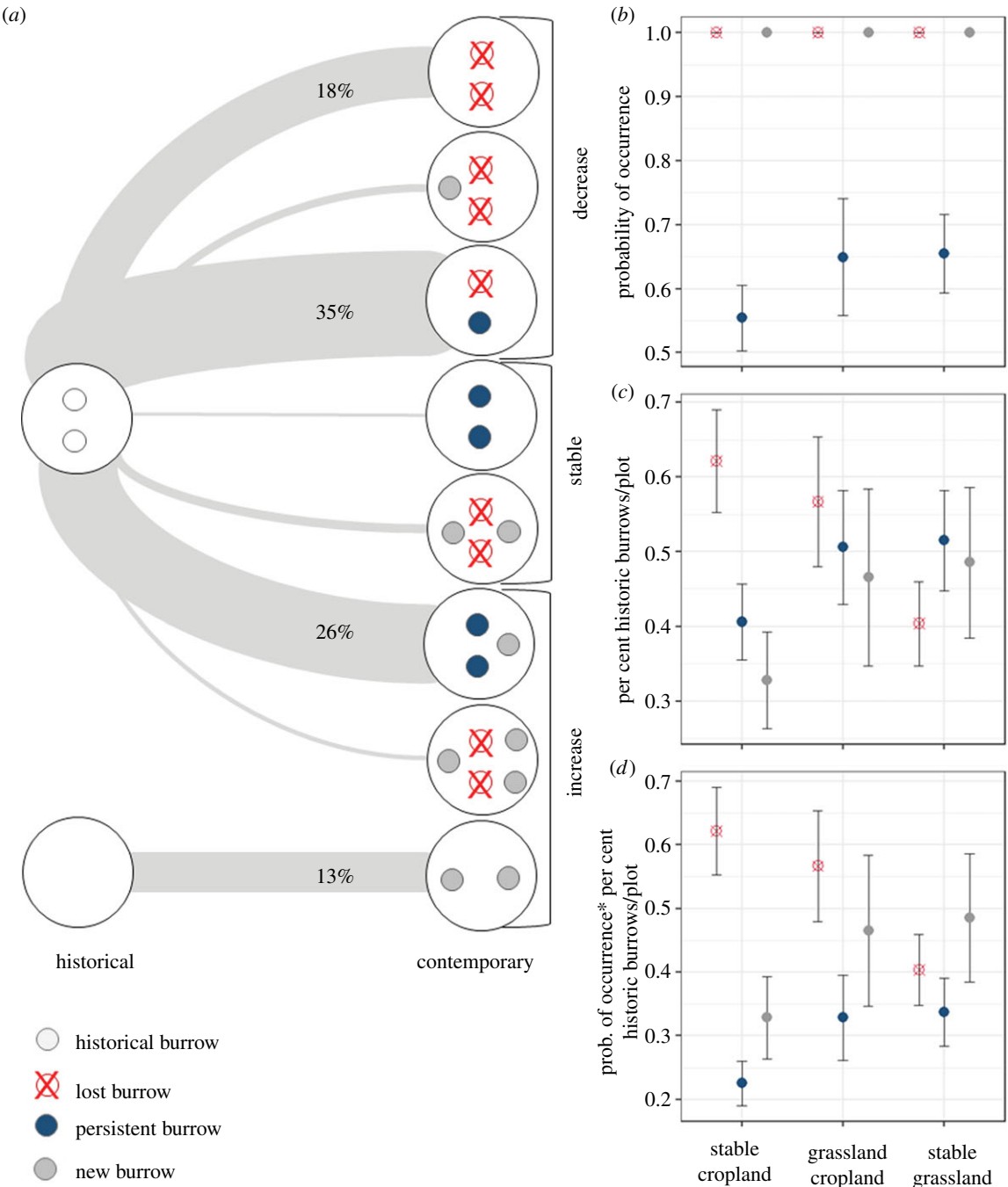

**Figure 4.** Observed change in burrow distribution over time. Per cent points on the lines indicate the percentage of plots following the respective trajectory. (*a*) Probability of occurrence of lost, persistent, and new burrows across three land-use change classes (zero-inflated part of the models). (*b*) The predicted proportion of lost, persistent, and new burrows per plot (conditional on the probability of occurrence) across three land-use change classes (count part of the models). (*c*) The mean proportion of burrows lost, persistent, and new burrows per plot. (*d*) All values represent predictions for a hypothetical plot starting out with 19 burrows (average value for occupied plots in the historical time period). (Online version in colour.)

to repeated and increased burrow disturbance and reduced food availability [23,49].

We showed that declines in burrow densities were steeper in persistent cropland compared to persistent grassland, and in plots that were cropped prior to the Virgin Lands Campaign, compared to plots converted later. The repeated disturbance of burrows through ploughing, likely led to increased colony stress and higher energy costs for re-establishing disturbed burrows [23,50,51], ultimately reducing colony fitness and size [23]. Because the declines were steepest in older fields, repeated disturbances associated with cropping may substantially decrease population size over time, despite the effects of single disturbances possibly being minor [22]. Additionally,

agricultural conversion likely reduced the forage quantity and quality for the marmots, which preferentially forage on natural vegetation [26,51]. It is likely that extensive agricultural practices—common in Kazakhstan until the early 2000s [10,40]—reduced forage availability during the fattening season, which in turn prevented marmots from gaining sufficient body mass to survive hibernation [23,52]. Taken together, the persistent cropping over 50 years, coupled with high rates of burrow disturbance and reduced forage availability may explain the observed long-term, gradual population decline. Although historical agricultural regimes could not be experimentally randomized across our study area, the Virgin Lands Campaign represented possibly the

largest natural experiment on the effects of agricultural conversion for biodiversity, and our results support the idea that an increase in the frequency of system disturbance can lead to long-term population declines [22].

In addition to cropland conversion, disease, poisoning, hunting, and trapping could have contributed to the gradual, long-term population decline we observed. Although it is possible that the effects of cropland conversion were locally modulated by these factors, effects of disease and poisoning are unlikely to be substantial at the spatial and temporal scale of our study. Parasites and disease may cause local mortality in marmot populations, but no major demographic effects have been reported for the bobak marmot in Central Asia since the mid-twentieth century [23]. Poisoning of burrowing mammals has been a common practice historically in parts of Canada, USA, and Mexico, but was not widely practiced in Kazakhstan [23,28]. Bobak marmot populations have been historically affected by overhunting and trapping, especially in Russia, but since the 1950s, hunting became regulated and the marmot population rebounded [26,50]. Furthermore, fur trapping and hunting are highest in proximity of human settlements, so their effects would be partially accounted for in our analyses via the predictor *distance to farm* [50].

An alternative explanation to the gradual, long-term decline we observe is that land-use intensification led to drops in marmot population due to indirect effects of pesticides and herbicides [24]. This drop, however, would have been recent and more abrupt, because intensification in Kazakhstan only started in the early 2000s, when over 2 million ha of cropland transitioned to no-till, and imports of herbicides increased substantially [10,40]. Pesticides and herbicides can affect marmots through direct contamination and by reducing forage availability during the fattening season [53]. Although preliminary field data suggested that marmot colonies have disappeared in some croplands where no-till (and thus heavy pesticide use) has been adopted, the average numbers of burrows per plot since 2000 did not change significantly (electronic supplementary material, S10), rendering intensification an unlikely explanation for the strong declines we found. However, systematic assessments of herbicide impacts over longer time periods would be beneficial to elucidate if and on which timescales pesticides affect population dynamics of burrowing mammals.

Despite the overall reduction in marmot burrows observed here, many individual marmot burrows persisted for approximately 50 years. This persistence is remarkable for a species with life expectancy ranging between 5 and 7 years [23]. A higher proportion of burrows retained their exact location in undisturbed grassland habitat compared to persistent croplands. Without disturbance, marmots tend to re-use the same wintering burrows for multiple years and spend less than 4 min per day maintaining their burrow [23], sometimes only changing the main entrance and the mound [51]—suggesting that our estimate of philopatry is likely conservative. The substantial past investment in burrow systems [23], combined with attractive early spring food availability from sprouting wheat [38] and potential competition for remaining suitable habitat may compel marmots to remain in suboptimal cropland habitat. However, because rates of burrow persistence were lower in cropland plots than in grassland plots, we suggest that philopatry in conjunction with the long-term agricultural use might create an ecological trap for the species in cropland fields

[30]. We caution that our study quantifies the persistence of burrows, not the philopatry of individuals themselves, but we expect these measures to be strongly correlated.

Last but not least, we found a higher probability of burrow occurrence in croplands, compared to grasslands both for the historical and the contemporary time period. High burrow densities in historical croplands, shortly after the end of the Virgin Lands Campaign, suggest that burrow numbers did not drop immediately following conversion, further offering evidence for a gradual, possibly time-delayed response to agricultural expansion [18,54]. Rather, the higher number of burrows in croplands likely reflects high burrow densities typical of grasslands prior to conversion. Furthermore, when emerging from hibernation in early spring, natural vegetation is still scarce, and sprouting wheat provides an attractive alternative resource [26,51]. This was suggested by a higher probability of occurrence in areas with high NDVI values during the period of wheat sprouting (electronic supplementary material, S4). Other potential explanations for the high densities in croplands include a correlation between the most suitable marmot habitat and the suitable conditions for 'agriculture, a process locally described as 'colonies absorbed by agriculture' [51]. Indeed, our modelling results suggested that loamy soils had higher burrow densities compared to clayey or stony soils, which are less favourable for agriculture [33] (electronic supplementary material, S4). Finally, we caution that our study could not differentiate between grazed and ungrazed steppes, both combined in our single 'grassland' class. Analyses of contemporary imagery suggest that burrow detection probability for ungrazed steppes (19%) is lower than for grazed steppes (46%) [33], which means that our estimates of burrow densities in grasslands may be conservative. Although our analyses could not account for detection bias statistically, because ground data were not available for the historical time period, we would expect detection rates between land uses to be similar across time periods. It is nevertheless possible that estimates for the historical time period are conservative, because the overall image quality is lower compared to recent imagery. This suggests that the estimated magnitude of the decline is also conservative.

Our work pioneers the broad-scale use of Corona imagery for ecology and conservation. This novel data source provides an opportunity to expand the analyses of landscape and population dynamics both in space and time [34,35,55]. Corona data have global coverage, up to 2 m ground resolution, and stereographic properties, making it suitable for a wide range of applications including the detection of burrows, anthills, mallee fowl nests, or individual trees. Although Corona imagery has been used for archaeological and geomorphological questions [35] and for forest extent analysis [34,36], it has to our knowledge never been used for ecology and conservation. We suggest that these data provide ample opportunities to better understand ecological processes both in regions with long land-use histories and in regions with relatively short land-use histories, where Corona data may provide a baseline for ecological assessments [13,38,45].

## 5. Conclusion

Most of the world's grasslands have been converted to agriculture to feed the growing human population, and continued expansion and intensification are threatening the biodiversity

of the remaining grassland ecosystems. The Eurasian steppes, some of the last large natural grasslands, have undergone major land-use changes in the past century, the biodiversity effects of which have not yet been fully quantified. Our broad-scale assessment of bobak marmot population changes highlights that ongoing population declines are likely related to agricultural expansion happening many decades ago—a trend that would have been overlooked without the long-term perspective facilitated by Corona imagery. Our work suggests that longitudinal assessments of population dynamics are essential for addressing current biodiversity challenges. Slow and possibly time-delayed responses will likely cause the full biodiversity effects of recent land-use changes to only become apparent several decades into the future. To safeguard the biodiversity of some of the most vulnerable ecosystems, conservation and management actions should consider long-term biodiversity responses to land conversions.

Data accessibility. Our research is based on publicly available remote sensing imagery (i.e. Google Earth, Bing). Historical Corona imagery for the study region is freely available via the United States Geological Survey Earth Explorer website (https://earthexplorer.usgs.gov/). Marmot burrow location and explanatory variables for 900 sample plots, model selection techniques, and detailed modelling results are included in the electronic supplementary material.

Authors' contributions. C.M., J.K., T.K., and D.M. designed the work, C.M., M.D.N., N.K., and B.M.K. analysed the data, C.M., J.K., A.K., D.M., B.M.K., and T.K. interpreted the results, A.V.P. provided data, C.M. wrote the manuscript, all authors substantively revised the manuscript.

Competing interests. We declare we have no competing interests.

Funding. This work was supported by the Marie Sklodowska-Curie Project EcoSpy (Grant Agreement 793554), the Volkswagen Foundation (project BALTRAK, #A112025), and the Transylvania University of Brasov, Romania. A.V.P. acknowledges EU FP7 ERA.Net Russia Plus (559 CLIMASTEPPE), the OFRC UrB RAS—Institute of Steppe (UrB RAS No. AAAA-A19-119080190044-5), and the DFF-Danish ERC Support Program (grant no. 116491, 9127-00001B).

Acknowledgements. We thank F. Poetzschner, M. Pratzer, and K. Kirchner, for digitizing historical data, M. Baumann for exploratory analyses, and R. Krämer and R. Urazaliyev for spatial data layers. We thank two reviewers and an associate editor for many valuable suggestions on how to improve this work.

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
