## [Reviewer comments · Proceedings of the Royal Society B: Biological Sciences]

Review History

RSPB-2019-2897.R0 (Original submission)

Review form: Reviewer 1

Recommendation

Major revision is needed (please make suggestions in comments)

Scientific importance: Is the manuscript an original and important contribution to its field?

Excellent

General interest: Is the paper of sufficient general interest?

Excellent

Quality of the paper: Is the overall quality of the paper suitable?

Excellent

Is the length of the paper justified?

Yes

Should the paper be seen by a specialist statistical reviewer?

No

Do you have any concerns about statistical analyses in this paper? If so, please specify them explicitly in your report.

Yes

It is a condition of publication that authors make their supporting data, code and materials available - either as supplementary material or hosted in an external repository. Please rate, if applicable, the supporting data on the following criteria.

Is it accessible?

Yes

Is it clear?

Yes

Is it adequate?

Yes

Do you have any ethical concerns with this paper?

No

Comments to the Author

This is an innovative and exciting study on the change in bobak marmot abundance in the steppe of Kazakhstan. The authors use satellite images from 1968/69 and 2002-2017 to record marmot burrows (white circular spots) in 900 circular plots with a diameter of 1 km, distributed across three regions in Northern Kazakhstan. The change in burrow density (as a proxy for marmot abundance) is related to land-use change, in particular the conversion of steppe grassland to cropland.

The manuscript is well written; the applied methods (use of US spy satellite data; application of zero-inflated negative binomial GLMM) are innovative and appropriate; the data base is sound; the results are exciting and should be important for a broad readership across the globe, including conservation biologists, steppe ecologists, researchers interested in historical ecology, remote sensing, or burrowing rodents. Therefore, this study certainly deserves to be published in RSPB.

However, in its current state, the manuscript has some shortcomings which need to be addressed by the authors before I can recommend to accept it for publication. In particular, I'm not convinced by the way the authors tell their story and how they present their results.

General comments

1.) The authors observed a marked decline in marmot burrow densities over the past 50 years and claim that this decline is a delayed response to the agricultural expansion in the mid-20th century (Virgin Lands Campaign). So far, I'm not convinced that this is really the case. I have the impression, the authors have a story and try to interpret their findings to fit to the story rather than to derive a new story from their results.

The loss of burrows was higher in plots with persistent agriculture since the 1960s compared to persistent grassland plots (lines 334-336). Thus, if conversion to cropland is really the cause of marmot decline, then the burrow loss between the surveyed periods could be due to the agricultural expansion before 1968. However, is agricultural expansion really the cause of the decline? So far, this is a fundamental assumption by the authors which stands in contrast to their own results: First, the authors find actually a higher density of burrows in cropland than in

grassland both in the historic and contemporary period. Thus cropland appears to be the preferred habitat for marmots. Second, the number of lost burrows is equally high in plots with persistent cropland and plots with change from grassland to cropland (Fig. 4C). Even in persistent grassland plots, burrow density has decreased. Thus, could there be another driver for the general decline in burrow density? I suggest that the authors discuss this issue in a less biased manner.

2.) The authors present their results in two ways. First, they give the model output (the regression coefficients) in Tables S3 and S4 in the supplementary material. These numbers are, however, difficult to interpret because they refer to a zero-inflated negative binomial model, which might not be familiar to the common reader. Second, they provide easy-to-interpret figures in the main text, such as declines in burrow numbers as percentage or Fig. 3. This approach is, in general, appropriate. However, here, it is not possible to tell how the figures given in the main text relate to the original model output. The reader has simply to believe that these figures are correct. I suggest that the authors provide some guidance in the supplementary material how the regression coefficients in Tables S3 and S4 can be interpreted, e.g. how they can be turned into predicted burrow counts for certain groups of plots such as 'historical plots in grasslands' or 'contemporary plots in cropland'. Since the interaction term (time*land use) is particularly important to understand the time-delayed decline in marmot densities, this term should be explained in more detail.

3.) The authors examine 900 plots for the occurrence and number of marmot burrows. They examine each plot twice, once for the period 1968/69 and once for the period 2002-2017. Thus, they have a repeated measures design. It is therefore absolutely necessary to include PlotID as a random effect in their GLMM. So far, data from the same plot in the 1960s and today are treated as independent.

4.) In lines 339-345 and Figure S3, the authors present an additional result that is based on additional data on the agricultural history for a subset of 111 plots. This result supports their main finding of a time-delayed marmot decline and is indeed very interesting. However, it remains obscure, what the source for the additional data is, why these additional data are not available for the other plots, and how these additional data were analysed to produce Fig. S3. I suggest, that the authors add this information to the Methods section or (at least) provide more details in the supplementary material.

5.) The authors use statements such as 'the declines [in marmot burrow densities] were steepest where cropland use persisted the longest' (lines 320-321) or 'the longer cropland persisted, the steeper the declines were in burrow numbers (line 331). These statements are misleading because they imply a gradual relationship between cropland age and burrow loss, where, in fact, there are only two categories ('persistent cropland' vs. 'grassland to cropland').

6.) In addition to land use type (grassland vs. cropland) or land use change (persistent cropland, persistent grassland, grassland-to-cropland), the authors use a large number of covariates, i.e. additional predictors in the models that shall be controlled, such as soil texture, NDVI, distance to farms etc. These explanatory variables are only very briefly introduced in the main text (267-288); Table S1 does not provide much more information. Furthermore, several covariates have significant effects in the models, but are not discussed at all. If space is really that limited in the main text, at least in the supplementary material the authors should describe each variable in more detail: How was it measured? For which time period? What is the used scale of measurement?

In particular it is unclear (a) how soil texture was quantified, (b) whether a contemporary NDVI can be used for the historical period as well, (c) how a 'river' is defined here, (d) whether climate change over a half century is indeed negligible, so far this is not convincing.

Specific comments

85-92: Not all readers will immediately understand why species populations respond delayed to habitat losses or changes. Could you briefly explain the mechanism? You say that socio-economic pressures might amplify the delay (l. 90-92). Could you give an example?

120: 'land-use change'?

141: 'burrow location' or 'the species' choice of burrow location'?

148: Here, you mention only Google Earth and Bing, but not ESRI which provided most of the data (line 214).

line 149, 195, 215 and Fig. 2 (caption and figure): Make time span for contemporary period consistent.

171-173: I do not really understand, why you refer to Fig. 2 here. In Fig. 2, I cannot see which plots or which proportion of plots is located in fallow or abandoned cropland. Is it possible to distinguish fallow or abandoned cropland from managed cropland or grassland on the satellite images? From the marmots' point of view, is an old field more similar to a cropland habitat or grassland habitat? Would a third category in Fig. 2 and in the analysis be helpful?

240: Do you mean plots when you say 'samples'? 622 plots out of 900 would be 69.1%, or 622 plots out of 1800 (900 per period) would be 34.6%. How do you get 36%? Ah, you excluded plots with cloud cover, did you not?

244-245: You mean 38 plots were located in fallow or abandoned fields in the contemporary period, but were actively cropped in 1968/69?

248: This sounds misleading. Rather say that you want to assess the temporal consistency in the spatial distribution of burrows.

253-255: What is the difference between 'probability of philopatry' and 'predicted number of persistent burrows per plot'? Is one term sufficient? I guess you divided the number of paired observations (times two?) by the total number of burrows across time periods? You should make this clear.

290-296: I guess, these AIC figures as well as the figures shown in Text S1 refer to your first (and main) model with the number of burrow counts as response. You should make this clear.

314-315: This is confusing. Above, you explained that land-use type and time are only predictors in the occurrence and density model, whereas in the models on maintained, lost and newly created burrows, land change is the main predictor. Thus, what do you mean with 'For all models' here?

320 + 384 + 428: Actually, from 1968 to 2018, it's 50 years not 60 years.

322-324: Leave this for the Discussion.

334-336: Are these simple observed average values? To which test does the p-value refer? Your occurrence and density model does not differentiate between persistent cropland plots and grassland-to-cropland plots. So where do these numbers come from? Moreover, in the following section (lines 363-365) you give different (but similar) numbers for the same thing... Is this redundancy needed?

402-404: This is not logical. Do you think that burrow densities in your historical grassland plots were untypically low?

Fig. 4A: The labels 'increase' and 'decrease' need to be exchanged.

Suppl. material, line 7: 'were available'

Suppl. material, line 65: This should be Table S4

Table S1: The 'www.' in the source URL for slope and NDVI need to be deleted.

Review form: Reviewer 2

Recommendation

Accept with minor revision (please list in comments)

Scientific importance: Is the manuscript an original and important contribution to its field?

Good

General interest: Is the paper of sufficient general interest?

Excellent

Quality of the paper: Is the overall quality of the paper suitable?

Excellent

Is the length of the paper justified?

Yes

Should the paper be seen by a specialist statistical reviewer?

No

Do you have any concerns about statistical analyses in this paper? If so, please specify them explicitly in your report.

No

It is a condition of publication that authors make their supporting data, code and materials available - either as supplementary material or hosted in an external repository. Please rate, if applicable, the supporting data on the following criteria.

Is it accessible?

N/A

Is it clear?

N/A

Is it adequate?

Yes

Do you have any ethical concerns with this paper?

No

Comments to the Author

The paper entitled "Cold War spy satellite images reveal delayed declines of a philopatric keystone species in response to cropland expansion" is well written, logical, and interesting. It is novel because the dataset of photos is not yet widely known and used by scientists. Statistical analyses are appropriate, as well as the iconographic part. I have some minor comments below

and a general comment about the accessibility of the Corona dataset.

Minor comments

1. 80 add a full stop after “[14-16]”
1. 88-89 rewrite this sentence “in mammals and birds” should be placed in earlier position
1. 280 add “km” after “17”
1. 325 the first sentence of this paragraph is partially a repetition of the paragraph before, please restructure the two paragraphs, or rephrase it to make clear that the first paragraph is a summary of the results

Accessibility of the Corona dataset

One of the most important novelty of this manuscript is the use of the database Corona. In the manuscript I did not find a permanent link to the USGS EarthExplorer where the photos can be downloadable. By the way, browsing the Corona dataset in the USGS Earth Explorer is not intuitive and the quality of the photos I have visualised is poor, and certainly below the resolution that they seem to have in Figure 1. This might be due to my not complete knowledge of the way to access the data, but I strongly recommend authors to: 1) clearly indicating a link and brief information on how accessing the aerial photos, and 2) verifying whether the resolution of the photos is reasonable also outside their study area. Due to space limitations, this information could be also in the supplementary materials.

Decision letter (RSPB-2019-2897.R0)

24-Feb-2020

Dear Dr Munteanu:

Your manuscript has now been peer reviewed and the reviews have been assessed by an Associate Editor. The reviewers' comments (not including confidential comments to the Editor) and the comments from the Associate Editor are included at the end of this email for your reference. As you will see, the reviewers and the Editors have raised some concerns with your manuscript and we would like to invite you to revise your manuscript to address them.

Research ethics:

Use of animals and field studies:

Please submit a copy of your revised paper within three weeks. If we do not hear from you

within this time your manuscript will be rejected. If you are unable to meet this deadline please let us know as soon as possible, as we may be able to grant a short extension.

Best wishes,
Dr Locke Rowe
mailto: proceedingsb@royalsociety.org

Associate Editor
Board Member: 1
Comments to Author:

Dear authors,

Two authors and myself have read your MS. We all like your MS and think it has the potential to be suitable for PRSB. Reviewer 2 only has minor comments, but reviewer 1 has very serious concerns about the interpretation of the results, which touch at the heart of this paper. Is the evidence from the results really that clear and correctly interpreted? I fully share these concerns, and these points need to be addressed in a careful and rigorous manner. In addition to the comments of reviewer 1, I wondered why the authors have only used spy satellite data from two points in time, and not data from years in between (or one or two equidistant years in between)? This would allow for a much more detailed insight in changes over time than comparing two distant points separated by 60 years, and could strengthen some of the interpretations that are currently in doubt. Overall, I therefore recommend a (major) revision.

Reviewer(s)' Comments to Author:

Referee: 1

Comments to the Author(s)

This is an innovative and exciting study on the change in bobak marmot abundance in the steppe of Kazakhstan. The authors use satellite images from 1968/69 and 2002-2017 to record marmot burrows (white circular spots) in 900 circular plots with a diameter of 1 km, distributed across three regions in Northern Kazakhstan. The change in burrow density (as a proxy for marmot abundance) is related to land-use change, in particular the conversion of steppe grassland to cropland.

The manuscript is well written; the applied methods (use of US spy satellite data; application of zero-inflated negative binomial GLMM) are innovative and appropriate; the data base is sound; the results are exciting and should be important for a broad readership across the globe, including conservation biologists, steppe ecologists, researchers interested in historical ecology, remote sensing, or burrowing rodents. Therefore, this study certainly deserves to be published in RSPB.

However, in its current state, the manuscript has some shortcomings which need to be addressed by the authors before I can recommend to accept it for publication. In particular, I'm not convinced by the way the authors tell their story and how they present their results.

General comments

1.) The authors observed a marked decline in marmot burrow densities over the past 50 years and claim that this decline is a delayed response to the agricultural expansion in the mid-20th century (Virgin Lands Campaign). So far, I'm not convinced that this is really the case. I have the impression, the authors have a story and try to interpret their findings to fit to the story rather than to derive a new story from their results.

The loss of burrows was higher in plots with persistent agriculture since the 1960s compared to persistent grassland plots (lines 334-336). Thus, if conversion to cropland is really the cause of marmot decline, then the burrow loss between the surveyed periods could be due to the agricultural expansion before 1968. However, is agricultural expansion really the cause of the decline? So far, this is a fundamental assumption by the authors which stands in contrast to their own results: First, the authors find actually a higher density of burrows in cropland than in grassland both in the historic and contemporary period. Thus cropland appears to be the preferred habitat for marmots. Second, the number of lost burrows is equally high in plots with persistent cropland and plots with change from grassland to cropland (Fig. 4C). Even in persistent grassland plots, burrow density has decreased. Thus, could there be another driver for the general decline in burrow density? I suggest that the authors discuss this issue in a less biased manner.

2.) The authors present their results in two ways. First, they give the model output (the regression coefficients) in Tables S3 and S4 in the supplementary material. These numbers are, however, difficult to interpret because they refer to a zero-inflated negative binomial model, which might not be familiar to the common reader. Second, they provide easy-to-interpret figures in the main text, such as declines in burrow numbers as percentage or Fig. 3. This approach is, in general, appropriate. However, here, it is not possible to tell how the figures given in the main text relate to the original model output. The reader has simply to believe that these figures are correct. I suggest that the authors provide some guidance in the supplementary material how the regression coefficients in Tables S3 and S4 can be interpreted, e.g. how they can be turned into predicted burrow counts for certain groups of plots such as 'historical plots in grasslands' or 'contemporary plots in cropland'. Since the interaction term (time*land use) is particularly important to understand the time-delayed decline in marmot densities, this term should be explained in more detail.

3.) The authors examine 900 plots for the occurrence and number of marmot burrows. They examine each plot twice, once for the period 1968/69 and once for the period 2002-2017. Thus, they have a repeated measures design. It is therefore absolutely necessary to include PlotID as a random effect in their GLMM. So far, data from the same plot in the 1960s and today are treated as independent.

4.) In lines 339-345 and Figure S3, the authors present an additional result that is based on additional data on the agricultural history for a subset of 111 plots. This result supports their main finding of a time-delayed marmot decline and is indeed very interesting. However, it remains obscure, what the source for the additional data is, why these additional data are not available for the other plots, and how these additional data were analysed to produce Fig. S3. I suggest, that the authors add this information to the Methods section or (at least) provide more details in the supplementary material.

5.) The authors use statements such as 'the declines [in marmot burrow densities] were steepest where cropland use persisted the longest' (lines 320-321) or 'the longer cropland persisted, the steeper the declines were in burrow numbers (line 331). These statements are misleading because they imply a gradual relationship between cropland age and burrow loss, where, in fact, there are only two categories ('persistent cropland' vs. 'grassland to cropland').

6.) In addition to land use type (grassland vs. cropland) or land use change (persistent cropland, persistent grassland, grassland-to-cropland), the authors use a large number of covariates, i.e. additional predictors in the models that shall be controlled, such as soil texture, NDVI, distance to farms etc. These explanatory variables are only very briefly introduced in the main text (267-288); Table S1 does not provide much more information. Furthermore, several covariates have significant effects in the models, but are not discussed at all. If space is really that limited in the main text, at least in the supplementary material the authors should describe each variable in more detail: How was it measured? For which time period? What is the used scale of measurement?

In particular it is unclear (a) how soil texture was quantified, (b) whether a contemporary NDVI can be used for the historical period as well, (c) how a 'river' is defined here, (d) whether climate change over a half century is indeed negligible, so far this is not convincing.

Specific comments

85-92: Not all readers will immediately understand why species populations respond delayed to habitat losses or changes. Could you briefly explain the mechanism? You say that socio-economic pressures might amplify the delay (l. 90-92). Could you give an example?

120: 'land-use change'?

141: 'burrow location' or 'the species' choice of burrow location'?

148: Here, you mention only Google Earth and Bing, but not ESRI which provided most of the data (line 214).

line 149, 195, 215 and Fig. 2 (caption and figure): Make time span for contemporary period consistent.

171-173: I do not really understand, why you refer to Fig. 2 here. In Fig. 2, I cannot see which plots or which proportion of plots is located in fallow or abandoned cropland. Is it possible to distinguish fallow or abandoned cropland from managed cropland or grassland on the satellite images? From the marmots' point of view, is an old field more similar to a cropland habitat or grassland habitat? Would a third category in Fig. 2 and in the analysis be helpful?

240: Do you mean plots when you say 'samples'? 622 plots out of 900 would be 69.1%, or 622 plots out of 1800 (900 per period) would be 34.6%. How do you get 36%? Ah, you excluded plots with cloud cover, did you not?

244-245: You mean 38 plots were located in fallow or abandoned fields in the contemporary period, but were actively cropped in 1968/69?

248: This sounds misleading. Rather say that you want to assess the temporal consistency in the spatial distribution of burrows.

253-255: What is the difference between 'probability of philopatry' and 'predicted number of persistent burrows per plot'? Is one term sufficient? I guess you divided the number of paired observations (times two?) by the total number of burrows across time periods? You should make this clear.

290-296: I guess, these AIC figures as well as the figures shown in Text S1 refer to your first (and main) model with the number of burrow counts as response. You should make this clear.

314-315: This is confusing. Above, you explained that land-use type and time are only predictors in the occurrence and density model, whereas in the models on maintained, lost and newly created burrows, land change is the main predictor. Thus, what do you mean with 'For all models' here?

320 + 384 + 428: Actually, from 1968 to 2018, it's 50 years not 60 years.

322-324: Leave this for the Discussion.

334-336: Are these simple observed average values? To which test does the p-value refer? Your occurrence and density model does not differentiate between persistent cropland plots and grassland-to-cropland plots. So where do these numbers come from? Moreover, in the following

section (lines 363-365) you give different (but similar) numbers for the same thing... Is this redundancy needed?

402-404: This is not logical. Do you think that burrow densities in your historical grassland plots were untypically low?

Fig. 4A: The labels 'increase' and 'decrease' need to be exchanged.

Suppl. material, line 7: 'were available'

Suppl. material, line 65: This should be Table S4

Table S1: The 'www.' in the source URL for slope and NDVI need to be deleted.

Referee: 2

Comments to the Author(s)

The paper entitled "Cold War spy satellite images reveal delayed declines of a philopatric keystone species in response to cropland expansion" is well written, logical, and interesting. It is novel because the dataset of photos is not yet widely known and used by scientists. Statistical analyses are appropriate, as well as the iconographic part. I have some minor comments below and a general comment about the accessibility of the Corona dataset.

Minor comments

l. 80 add a full stop after "[14-16]"

l. 88-89 rewrite this sentence "in mammals and birds" should be placed in earlier position

l. 280 add "km" after "17"

l. 325 the first sentence of this paragraph is partially a repetition of the paragraph before, please restructure the two paragraphs, or rephrase it to make clear that the first paragraph is a summary of the results

Accessibility of the Corona dataset

One of the most important novelty of this manuscript is the use of the database Corona. In the manuscript I did not find a permanent link to the USGS EarthExplorer where the photos can be downloadable. By the way, browsing the Corona dataset in the USGS Earth Explorer is not intuitive and the quality of the photos I have visualised is poor, and certainly below the resolution that they seem to have in Figure 1. This might be due to my not complete knowledge of the way to access the data, but I strongly recommend authors to: 1) clearly indicating a link and brief information on how accessing the aerial photos, and 2) verifying whether the resolution of the photos is reasonable also outside their study area. Due to space limitations, this information could be also in the supplementary materials.

Author's Response to Decision Letter for (RSPB-2019-2897.R0)

See Appendix A.

RSPB-2019-2897.R1 (Revision)

Review form: Reviewer 1

Recommendation

Major revision is needed (please make suggestions in comments)

Scientific importance: Is the manuscript an original and important contribution to its field?

Excellent

General interest: Is the paper of sufficient general interest?

Excellent

Quality of the paper: Is the overall quality of the paper suitable?

Excellent

Is the length of the paper justified?

Yes

Should the paper be seen by a specialist statistical reviewer?

No

Do you have any concerns about statistical analyses in this paper? If so, please specify them explicitly in your report.

Yes

It is a condition of publication that authors make their supporting data, code and materials available - either as supplementary material or hosted in an external repository. Please rate, if applicable, the supporting data on the following criteria.

Is it accessible?

Yes

Is it clear?

Yes

Is it adequate?

Yes

Do you have any ethical concerns with this paper?

No

Comments to the Author

The authors have substantially revised their manuscript, including the statistical analyses. They have also added some extra analyses. The revised and new results strengthen the previous conclusions of the authors. Most important, the decline in marmot burrows is now restricted to cropland, while in grassland, the expected number of burrows even increased. The revised Figs. 3 and 4 are better to interpret now, in particular together with explanations provided in the new Supplementary Material 11, which is very helpful. The provided explanations for the counterintuitive higher burrow densities in croplands compared to grasslands are reasonable. Overall, the manuscript has gained much in clarity.

Although the authors' line of thought is more convincing now, I still have some doubts concerning the assumed delayed response of the marmot population to agricultural expansion

(see first and third comment below). Moreover, the clarity of the revised manuscript has opened my eyes for another shortcoming of the statistical analysis (comment 2). However, I would like to stress that, in my opinion, a re-revised version of this manuscript would make a great contribution to RSPB.

1. The authors assume that the decline in marmot burrows on cropland over the last 50 years is a delayed response to the conversion of steppe grassland to cropland during the Virgin Lands Campaign (1954-1963). However, the mechanism by which agricultural land-use caused declines in marmot populations remains obscure in the manuscript for a long time. This is not at all clear given that cropland features a much higher burrow density than grassland (Fig. 3). In contrast, the authors stress the good food resources on cropland (l. 364-371, 416-417). Only in the second half of the Discussion, the authors explain, why they assume detrimental effects of land conversion on marmot populations (unfortunately without citing any reference): 'The repeated disturbance of the burrows through agricultural practices (i.e. tillage, harvest, pesticide application) might lead to population fitness declines, ultimately causing a population drop (418-420).' I suggest to move this part to the Introduction, to elaborate a little bit more on this and include other evidence for negative effects of agricultural practices on marmots. This would make the reader less doubtful on the authors' assumption.

2. The authors observed a remarkable philopatric behaviour in bobak marmots. Moreover they found 'a higher proportion of burrows retaining their exact location in undisturbed grassland habitat than in croplands' (l. 348-349). This is used as an argument to support their assumption that agricultural practices lead to the long-term population decline (l. 409-422). Although this conclusion is probably correct, the authors did not formally test, whether the proportion of persistent burrows is higher in grassland than in cropland. What they tested is whether the absolute number of persistent burrows is higher in grassland than in cropland, which is not the case (Fig. 4). The same is true for the number of lost and gained burrows. Since the number of burrows is generally higher in cropland than in grassland, these tests are misleading. I suggest, that the authors use the proportion of persistent, lost and gained burrows as response variable in their models, instead of the absolute numbers.

3. The authors mention that 'low-intensity agricultural practices' were 'historically common in Kazakhstan' (l. 369-370). Does this mean that agriculture has been intensified during the last 50 years (at least on the most suitable sites where it was not abandoned)? Could this be the reason for the marmot decline?

4. The language of added or revised text sections (main text and supplementary material) does not meet the standard of the journal and needs editing.

Specific comments

1-2: The original title was shorter and better.

84-87: Socio-economic pressures may result in land conversions and associated habitat loss or habitat degradation, which in turn lead to local species extinctions. But why do species respond delayed and not immediately? This cannot be explained with socio-economic pressures but only with the species' ecology, e.g. traits like longevity, philopatric behaviour, slow population dynamics.... Please, provide one example, at best for a mammal.

131-132: This final expectation is not comprehensible. Better leave as in the original manuscript.

134-135: "marmot burrows, and related them to the surrounding"

155-156: "this campaign"

271-278: The terminology and the order of terms (maintained, lost, newly created) differ from those in l. 229-232 (lost, gained, persistent). Be consistent throughout the manuscript and the supplementary material.

302-305: Omit these general statements without any reference. They are redundant anyway.

305: It would be interesting to get to know the average burrow densities in both time periods. The percent decrease can then be set in parentheses. Moreover, given the large range in decreases or increases from -60 to +55 burrows/plot, an average decrease be 14% might not be significant. I

guess you have tested this decrease independent of land-use with a more simple GLMM?

312: Where does this 43% come from? According to Fig. 3B the decline is about 30% and according to Fig. 3C the decline is about 60% (as you write in line 317).

322-323: To which 'pattern' do you refer here? So far you have not described any pattern that could be confirmed. What you describe in Supplementary Material 9 is the pattern itself.

329-330: The 'massive agricultural expansion' occurred before the historical time period and not between time periods. Do not confuse the reader.

Suppl. Mat. 9: You analysed 36 preVLC and 95 VLC plots (together 131), but talk about a total of 111 plots for this analysis?

Fig. 3 + 4: Refer in the caption to Suppl. Mat. 11.

Suppl. Mat. 9: Have you tested the decline in mean burrow numbers statistically? How?

Decision letter (RSPB-2019-2897.R1)

25-Mar-2020

Dear Dr Munteanu:

Your manuscript has now been peer reviewed and the reviews have been assessed by an Associate Editor. The reviewers' comments (not including confidential comments to the Editor) and the comments from the Associate Editor are included at the end of this email for your reference. As you will see, the reviewers and the Editors have raised some concerns with your manuscript and we would like to invite you to revise your manuscript to address them.

We urge you to make every effort to fully address all of the comments in this final revision. If deemed necessary by the Associate Editor, your manuscript will be sent back to one or more of the original reviewers for assessment. If the original reviewers are not available we may invite new reviewers. Please note that we cannot guarantee eventual acceptance of your manuscript at this stage.

Research ethics:

Use of animals and field studies:

Please submit a copy of your revised paper within three weeks. If we do not hear from you within this time your manuscript will be rejected. If you are unable to meet this deadline please let us know as soon as possible, as we may be able to grant a short extension.

Best wishes,
Dr Locke Rowe
Editor, Proceedings B
mailto: proceedingsb@royalsociety.org

Associate Editor

Board Member: 1

Comments to Author:

Dear authors,

The most critical previous reviewer and myself have read the revision in detail. We think it has improved in clarity, but still think that quite some more work is needed. Yes, the topic is very suitable for the journal, we are all enthusiastic about the design of the study, very nice long-term dataset and novel use of spy satellite data, but the interpretation of results and Discussion is still not the most insightful and lacks critical dissemination.

Specifically, the reviewer highlights that the discussion provides very little insights into the mechanism underlying the differential temporal responses in grass and cropland. In addition, I also am totally not convinced about the strong emphasis in the MS for a time-delay in the response and how this is inferred from the analyses. The reviewer and myself also have a 1-2 other major, but otherwise mostly minor comments that also need to be addressed, and the text needs to be thoroughly checked for language (we have only highlighted some of the errors we encountered).

Overall, I am quite ambivalent about what to do with this MS. Both reviewers liked the MS very much and emphasized that it is important to be published, but at the same time the revision struggles to get the Discussion up to standard for our journal in my opinion. The work that needs to be done clearly falls in the major revision category as it touches on the key message of the MS, even if it may mainly concern rewriting of some parts of the Discussion. Technically, journal policy actually does not allow for a second round of revision. Notwithstanding, I recommend to the senior editor to seriously consider allowing for another round of major revision in this specific case, but I can see it go either way.

Major comment AE:

1. The discussion starts by stating “We reveal one of the longest recorded time-delayed response of a mammal to agricultural conversion (up to 50 years)” The “time-delay” in the response is not mentioned in the Results at all, so it comes out of the blue here as one of the main conclusions of the MS. The results show that declines were most prominent in persistent croplands and in those plots where cropland use persisted the longest. The discussion should then first explain why this can be interpreted as a delayed response and what the caveats are. By interpreting the stronger decline in persistent croplands compared to more recent croplands to a time-delayed response, the authors interpret this as a time delayed effect. But croplands converted earlier and later may differ in many ways (the authors suggest that they converted the best lands first), and it is unlikely that all of the factors may have been accounted for in the models (the agricultural regimes were not experimentally randomized treatments, which limits the inference that can be drawn, which should be acknowledged in the MS).

Also how does the delay work mechanistically is not really explained. In the intro it is suggested that ‘Land conversions lead not only to instantaneous, but also time-delayed responses in species richness, distribution, diversity and abundance.’ But there may also be difference in changes over time in intensity of land use between these type of croplands (I am not an expert on farming in Kazakhstan, but increased plowing, pesticides, earlier converted may have depleted the resources earlier thereby increasing needs for fertilizer). I do not necessarily doubt the interpretation of the authors, just that the delayed response is presented as a given with very little critical discussion and suggestion for mechanisms.

L383 here you write ‘if marmots respond with a time-delay to agricultural conversion’ It seems the authors themselves also think here that the conclusion of a time delay is not strongly supported yet. In addition, testing for a time delay was also not mentioned as a study aim at the end of the Introduction, which suggest to me the study did not set out to look into this. Why is it then so prominently emphasized (e.g. in title of MS)?

2. L208. I was quite surprised to read here about this other paper [ref 48] by some of the same authors on using satellite imagery on the same species and location on a very similar questions. Why is this paper not mentioned in the Introduction (and cover letter for that matter), and explained how the current MS advances or is different from the previous work? Also the MS

now suggests there were no false positives in the validation study, but ref 48 states the opposite.

Most importantly, ref 48 suggest the detection bias depends on land use (their fig. 3 below). There seems to be a very large difference in underestimation from satellite data for example between grazed (GR) and ungrazed (UN) grasslands. The current study does not distinguish grazed and ungrazed grassland and bins them and studies burrow densities using satellite data over time. At the same time the authors state that grazing has increased over time. So can land use -dependne t detection and land use change within grassland and cropland affect the results on changes in burrow densities over time? Finally, in the validation a critical (but untested) assumption is made that the deterction bias patterns from is the same in recent and cold war satellite data. Clearly more critical discussion is needed.

Also L302: Do these numbers account for imperfect detection and differences in detection probability among habtiat? L333 How is this analyses affected by incomplete detection?

Minor comments:

L132 relocation. Reword? You do not follow individual marmots/burrows over time, so if a burrow disspeared you do not now whether it relocated or went extinct.

L155 his->the

L160 add comma after Union

L180 To me the usage of 'acquired' suggest thre photograph were bought in 1968, do you mean 'taken'

L296 delete "were"

L307 what model are the authors referring to here? L310 "predicted', do you mean the estimated difference between habitats from the data while accounting for other confounding effects in the statistical model? Predicted suggest to me this was an expectation, but it is in fact a result if I understand it correctly.

L327 inconsistent use of abbreviation VLC

L329 marmots-> you do not analyze marmots, but marmot burrows. Equating burrow philopatry with marmot philopatry is more something for the Discussion I suggest.

L347 burrow philopatry

Reviewer(s)' Comments to Author:

Referee: 1

Comments to the Author(s)

The authors have substantially revised their manuscript, including the statistical analyses. They have also added some extra analyses. The revised and new results strengthen the previous conclusions of the authors. Most important, the decline in marmot burrows is now restricted to cropland, while in grassland, the expected number of burrows even increased. The revised Figs. 3 and 4 are better to interpret now, in particular together with explanations provided in the new Supplementary Material 11, which is very helpful. The provided explanations for the counterintuitive higher burrow densities in croplands compared to grasslands are reasonable. Overall, the manuscript has gained much in clarity.

Although the authors' line of thought is more convincing now, I still have some doubts concerning the assumed delayed response of the marmot population to agricultural expansion (see first and third comment below). Moreover, the clarity of the revised manuscript has opened my eyes for another shortcoming of the statistical analysis (comment 2). However, I would like to stress that, in my opinion, a re-revised version of this manuscript would make a great contribution to RSPB.

1. The authors assume that the decline in marmot burrows on cropland over the last 50 years is a delayed response to the conversion of steppe grassland to cropland during the Virgin Lands

Campaign (1954-1963). However, the mechanism by which agricultural land-use caused declines in marmot populations remains obscure in the manuscript for a long time. This is not at all clear given that cropland features a much higher burrow density than grassland (Fig. 3). In contrast, the authors stress the good food resources on cropland (l. 364-371, 416-417). Only in the second half of the Discussion, the authors explain, why they assume detrimental effects of land conversion on marmot populations (unfortunately without citing any reference): 'The repeated disturbance of the burrows through agricultural practices (i.e. tillage, harvest, pesticide application) might lead to population fitness declines, ultimately causing a population drop (418-420).' I suggest to move this part to the Introduction, to elaborate a little bit more on this and include other evidence for negative effects of agricultural practices on marmots. This would make the reader less doubtful on the authors' assumption.

2. The authors observed a remarkable philopatric behaviour in bobak marmots. Moreover they found 'a higher proportion of burrows retaining their exact location in undisturbed grassland habitat than in croplands' (l. 348-349). This is used as an argument to support their assumption that agricultural practices lead to the long-term population decline (l. 409-422). Although this conclusion is probably correct, the authors did not formally test, whether the proportion of persistent burrows is higher in grassland than in cropland. What they tested is whether the absolute number of persistent burrows is higher in grassland than in cropland, which is not the case (Fig. 4). The same is true for the number of lost and gained burrows. Since the number of burrows is generally higher in cropland than in grassland, these tests are misleading. I suggest, that the authors use the proportion of persistent, lost and gained burrows as response variable in their models, instead of the absolute numbers.

3. The authors mention that 'low-intensity agricultural practices' were 'historically common in Kazakhstan' (l. 369-370). Does this mean that agriculture has been intensified during the last 50 years (at least on the most suitable sites where it was not abandoned)? Could this be the reason for the marmot decline?

4. The language of added or revised text sections (main text and supplementary material) does not meet the standard of the journal and needs editing.

Specific comments

1-2: The original title was shorter and better.

84-87: Socio-economic pressures may result in land conversions and associated habitat loss or habitat degradation, which in turn lead to local species extinctions. But why do species respond delayed and not immediately? This cannot be explained with socio-economic pressures but only with the species' ecology, e.g. traits like longevity, philopatric behaviour, slow population dynamics.... Please, provide one example, at best for a mammal.

131-132: This final expectation is not comprehensible. Better leave as in the original manuscript.

134-135: "marmot burrows, and related them to the surrounding"

155-156: "this campaign"

271-278: The terminology and the order of terms (maintained, lost, newly created) differ from those in l. 229-232 (lost, gained, persistent). Be consistent throughout the manuscript and the supplementary material.

302-305: Omit these general statements without any reference. They are redundant anyway.

305: It would be interesting to get to know the average burrow densities in both time periods. The percent decrease can then be set in parentheses. Moreover, given the large range in decreases or increases from -60 to +55 burrows/plot, an average decrease be 14% might not be significant. I guess you have tested this decrease independent of land-use with a more simple GLMM?

312: Where does this 43% come from? According to Fig. 3B the decline is about 30% and according to Fig. 3C the decline is about 60% (as you write in line 317).

322-323: To which 'pattern' do you refer here? So far you have not described any pattern that could be confirmed. What you describe in Supplementary Material 9 is the pattern itself.

329-330: The 'massive agricultural expansion' occurred before the historical time period and not between time periods. Do not confuse the reader.

Suppl. Mat. 9: You analysed 36 preVLC and 95 VLC plots (together 131), but talk about a total of 111 plots for this analysis?

Fig. 3 + 4: Refer in the caption to Suppl. Mat. 11.

Suppl. Mat. 9: Have you tested the decline in mean burrow numbers statistically? How?

Author's Response to Decision Letter for (RSPB-2019-2897.R1)

See Appendix B.

RSPB-2019-2897.R2 (Revision)

Review form: Reviewer 1

Recommendation

Accept with minor revision (please list in comments)

Scientific importance: Is the manuscript an original and important contribution to its field?

Excellent

General interest: Is the paper of sufficient general interest?

Excellent

Quality of the paper: Is the overall quality of the paper suitable?

Excellent

Is the length of the paper justified?

Yes

Should the paper be seen by a specialist statistical reviewer?

No

Do you have any concerns about statistical analyses in this paper? If so, please specify them explicitly in your report.

No

It is a condition of publication that authors make their supporting data, code and materials available - either as supplementary material or hosted in an external repository. Please rate, if applicable, the supporting data on the following criteria.

Is it accessible?

Yes

Is it clear?

Yes

Is it adequate?

Yes

Do you have any ethical concerns with this paper?

No

Comments to the Author

The authors have done a great job in revising their interpretation of results and the Discussion. The line of thought is much more convincing now. The revision of their statistical analyses has once more strengthened their findings. The authors have considered all my comments and I agree with the changes made. I have no further general comments and recommend to accept this manuscript for publication in Proceedings B.

Minor comments

l. 84: Keep either richness or diversity.

l. 231: "number of persistent burrows per plot"

Decision letter (RSPB-2019-2897.R2)

24-Apr-2020

Dear Dr Munteanu

I am pleased to inform you that your manuscript entitled "Cold War spy satellite images reveal long-term declines of a philopatric keystone species in response to cropland expansion." has been accepted for publication in Proceedings B.

The referee did have two word change suggestions, which could easily be dealt with when checking the proofs.

l. 84: Keep either richness or diversity.

l. 231: "number of persistent burrows per plot"

Open Access

Paper charges

Sincerely,

Dr Locke Rowe
Editor, Proceedings B
mailto:proceedingsb@royalsociety.org

Associate Editor:
Board Member: 1
Comments to Author:
(There are no comments.)

Board Member: 2
Comments to Author:
(There are no comments.)

Appendix A

Response to Editor and Reviewers

Comments by Editor:

We all like your MS and think it has the potential to be suitable for PRSB. Reviewer 2 only has minor comments, but reviewer 1 has very serious concerns about the interpretation of the results, which touch at the heart of this paper. Is the evidence from the results really that clear and correctly interpreted? I fully share these concerns, and these points need to be addressed in a careful and rigorous manner. In addition to the comments of reviewer 1, I wondered why the authors have only used spy satellite data from two points in time, and not data from years in between (or one or two equidistant years in between)? This would allow for a much more detailed insight in changes over time than comparing two distant points separated by 60 years, and could strengthen some of the interpretations that are currently in doubt. Overall, I therefore recommend a (major) revision.

Response:

Thank you for the very helpful and constructive reviews of our paper, as well as for the opportunity to revise and resubmit the manuscript. The comments of the reviewers and editor and included great suggestions on how to improve the manuscript. Allow us to briefly summarize our review here, and please see responses and revisions in the detailed section below, as well as in the manuscript.

We apologize for not thoroughly interpreting our results in the initial submission. To address the concern of the Reviewer 1 and the Editor, we have implemented several major revisions and additional analyses to strengthen our manuscript.

First, we have revised our statistical analyses (by including plot as a random effect, as suggested by Reviewer 1), reran the entire analyses, and more rigorously interpreted these new results. The revised results continue to highlight strong declines in burrow number in persistent cropland plots but also suggest small increases in persistent grassland plots. We have also included a new section on the interpretation of the statistical results (i.e., translation of the model coefficients into predicted burrow densities). Particularly, please see revised Fig 3 and Fig 4 as well as the revised Results section Line 314 onwards.

Second, to address the Editor's and Reviewer 1's suggestion of expanding the analyses to further points in time, we have expanded our supporting analyses to one additional point in time in the 1950s and four 5-year time intervals since 2000. The revised manuscript includes a new section in the Methods, describing these additional analyses, as well as new sections in the Supplementary Material providing further details on the datasets used and the analyzed carried out.

Unfortunately, we could not include one additional time point from spy satellite data as there are not enough images of high quality for this. However, in the revised manuscript we discussed about spatio-temporal limitations of the Corona data clearer and, following the suggestion of Reviewer 2, we included a new a section on Corona data availability and accessibility. For details, please see Supplementary Material 1.

Finally, we thoroughly revised the Discussion section in light of the Editors' and Reviewer 1's very helpful comments, ensuring that our results are objectively presented according to our analyses and results. We also included additional information to the Discussion about other possible processes that may affect the decline of burrow densities such as hunting or disease. Please see our fully revised Discussion section line 345 onwards.

We very much appreciate the thorough and thoughtful comments, which helped us to improve our manuscript. Below, we address each comment in more detail.

Comments by Reviewer 1:

The manuscript is well written; the applied methods (use of US spy satellite data; application of zero-inflated negative binomial GLMM) are innovative and appropriate; the data base is sound; the results are exciting and should be important for a broad readership across the globe, including conservation biologists, steppe ecologists, researchers interested in historical ecology, remote sensing, or burrowing rodents. Therefore, this study certainly deserves to be published in RSPB. However, in its current state, the manuscript has some shortcomings which need to be addressed by the authors before I can recommend to accept it for publication. In particular, I'm not convinced by the way the authors tell their story and how they present their results.

Thank you for your very thorough and constructive review, valuable feedback and time spent on supporting us to improve our manuscript. You are of course right that there are likely a number of factors at play driving the changes in marmot population sizes. In hindsight, we realize that we have not discussed them in sufficient detail, and neither did we discuss how they relate to our findings. We apologize for this shortcoming. In the revised manuscript, we have given more thorough consideration to other potential factors of marmot decline. We have revised the Discussion section in its entirety, added further relevant references to the text, and interpreted the (revised) results in light of these findings.

In addition, as suggested by the reviewer, we have improved our statistical analyses by including plot random effects in our analyses, added information on how model coefficients relate to the predicted values from revised Figures 3 and 4, included additional information on co-variates and their interpretation as well as expanded the methods section to include the analyses of additional data sources for multiple-points in time. Please see our detailed responses below.

We are immensely grateful for all your suggestions; and entrust you will find the improvements we brought to the manuscript acceptable.

General comments by Reviewer 1:

1.) The authors observed a marked decline in marmot burrow densities over the past 50 years and claim that this decline is a delayed response to the agricultural expansion in the mid-20th century (Virgin Lands Campaign). So far, I'm not convinced that this is really the case. I have the

impression, the authors have a story and try to interpret their findings to fit to the story rather than to derive a new story from their results.

The loss of burrows was higher in plots with persistent agriculture since the 1960s compared to persistent grassland plots (lines 334-336). Thus, if conversion to cropland is really the cause of marmot decline, then the burrow loss between the surveyed periods could be due to the agricultural expansion before 1968. However, is agricultural expansion really the cause of the decline? So far, this is a fundamental assumption by the authors which stands in contrast to their own results: First, the authors find actually a higher density of burrows in cropland than in grassland both in the historic and contemporary period. Thus cropland appears to be the preferred habitat for marmots. Second, the number of lost burrows is equally high in plots with persistent cropland and plots with change from grassland to cropland (Fig. 4C). Even in persistent grassland plots, burrow density has decreased. Thus, could there be another driver for the general decline in burrow density? I suggest that the authors discuss this issue in a less biased manner.

Thank you for raising a number of valuable points here. We address each of these below:

First, we apologize for the language used in our initial submission. In hindsight, we see how it has come across as “biased”. We have thoroughly revised our language to remove expressions like “confirmed our expectation”. We have also revised the title of the manuscript to read “Cold War spy satellite images reveal substantial, but delayed declines of a philopatric keystone species following cropland expansion”

Second, following your suggestion no 2 (below) we have revised our models and results (which now include random effects for plots as well). Please see the revised Fig 3 and Fig 4 as well as the corresponding revised results (Line 314 onwards). In light of these revisions, we now observe a small increase in burrow numbers in grassland plots. This finding suggests that despite high occurrence probability in croplands, burrow numbers have decreased strongest in cropland, and increased slightly in grasslands – pointing to potential negative effects of persisting cropping.

Third, we have revised and expanded our interpretation of higher densities in croplands compared to grasslands. Here we would like to highlight several potential explanations, including overlap of suitable agricultural areas with (most) suitable marmot habitat, the existence of a high number of “protection” burrows (maintained by marmots in areas with high predation or disturbance risk) in cropland areas and a possible legacy effect of the Virgin Lands Campaign (i.e., densities in cropland still reflecting the habitat situation from prior to croplands being converted by plowing virgin steppes). Please see revised manuscript at:

Line 357: “Potential explanations for the high densities in croplands include a correlation between the most suitable marmot habitat and the suitable conditions for agricultural, as well as the feeding of marmots on cropland resources. The expansion of the Virgin Lands Campaign in the mid 20th century affected areas with the highest soil quality — areas also preferred by the marmots for the ease of digging. This is a process locally known as ‘colonies absorbed by agriculture’ [53]. Indeed, our modelling results suggested that loamy soils had higher burrow densities compared to clayey or stony soils, which are less favorable for agriculture [48] (Supplementary Material 4). Although bobak marmots forage preferentially on natural vegetation [26,53], when emerging from hibernation in early spring, natural vegetation is still scarce, and the sprouting wheat may provide an attractive food source for the species. This explanation is

supported by our data, which indicates a higher probability of occurrence in areas with high NDVI values specifically in the month of May when wheat typically sprouts (Supplementary Material 4). Furthermore, low-intensity agricultural practices, historically common in Kazakhstan, likely provided some access to non-crop plants in crop fields, even during the fattening season. Finally, the higher burrow density in croplands may arise from a higher number of ‘protection’ burrows, often dug in areas where disturbance or predation is high [26,48]. Taken together, these elements may provide explanations for the higher probability of occurrence of burrows in cropland areas.”

Line 380: “The high densities in historical cropland plots, may also partially still reflect higher densities typical for grasslands, because many of these plots were likely converted from steppe to agriculture during the Virgin Lands Campaign, only a few years prior to our data collection. If marmots respond with a time-delay to agricultural conversion, then burrow densities which were particularly high in historical croplands (compared to historical grasslands), may still reflect densities typical for grasslands prior to conversion.”

Fourth, we apologize for not giving sufficient attention to other potential explanations for the observed patterns. In the revised manuscript, we discuss other factors that may affect declines in populations such as hunting, persecution or disease. We also discuss how these aspects may influence our findings.

Please see revised Discussion at:

Line 423: “Our results support prior evidence indicating that bobak population declines especially in Ukraine, Russia and Kazakhstan, are related to habitat conversion [26,31,48]. Declines induced by agricultural conversion, may be further modulated by hunting and disease (which we could not account for in our analyses). Although parasites and disease may cause local mortality in marmot populations, no major demographic effects have been reported for the bobak marmot [31]. Overhunting might have caused declines in populations in the early 20th century but hunting bags decreased strongly after the break- up of the Soviet Union [48,56,57] and have not recovered. Although hunting and disease may modulate the change observed in the population at large, we would not expect the effects to vary significantly between cropland and grassland habitat. Moreover, we would not expect hunting to affect the rate of philopatry in different land uses.”

*2.) The authors present their results in two ways. First, they give the model output (the regression coefficients) in Tables S3 and S4 in the supplementary material. These numbers are, however, difficult to interpret because they refer to a zero-inflated negative binomial model, which might not be familiar to the common reader. Second, they provide easy-to-interpret figures in the main text, such as declines in burrow numbers as percentage or Fig. 3. This approach is, in general, appropriate. However, here, it is not possible to tell how the figures given in the main text relate to the original model output. The reader has simply to believe that these figures are correct. I suggest that the authors provide some guidance in the supplementary material how the regression coefficients in Tables S3 and S4 can be interpreted, e.g. how they can be turned into predicted burrow counts for certain groups of plots such as ‘historical plots in grasslands’ or ‘contemporary plots in cropland’. Since the interaction term (time*land use) is particularly*

important to understand the time-delayed decline in marmot densities, this term should be explained in more detail.

Great point! To address your comment, we have made three major revisions. **First**, we have revised Figure 3 and Figure 4 to include three elements: a) the probability of burrow occurrence ($P(y>0)$), b) the predicted number of burrows, given burrows are present ($E(y|y>0)$) and c) the expected average number of burrows per plot ($E(y)$). Please see the revised figures and figure captions as well as Supplementary Material 11 for formulae.

Second, we clarified what the values in Figure 3 and Figure 4 represent: “We estimated the effects of land-use classes and time on the probability of occurrence and on burrow density, while keeping all other variables at their mean values. (Supplementary Material 6)”. **Third**, and maybe most importantly, we wrote an entirely new section, Supplementary Material 11, which includes details on the model fitting, the relation between the coefficients and the prediction, as well as the formula used for plotting Fig. 3 and Fig 4. This section reads as follows:

“Supplementary Material 11: Interpretation of model results

To predict probability of occurrence and expected number of burrows, we fixed all variables to their mean except the variables *land use* and *time period*. We estimated the random effect of the plot and the zone at the population average (assumed to be zero). In other words, we substituted the assumed population mean for an unknown random effect (also called marginal prediction). We used the package ‘*glmmTMB*’ in R to fit the model and predict values to a new dataset [11,12]. The models used consisted of two parts: the zero component of the model yields regression coefficients on the probability of additional zeros. Coefficients were modelled using a logit link and their interpretation is analogue to a logit model (i.e. by calculating the exponential of the model coefficients) but reflecting the chance of observing additional zeros rather than the probability of occurrence. The probability of occurrence can be calculated from the zero-inflated density using the formula:

$$P(y_i > 0) = (1 - \pi_i) * (1 - (\delta_i / (\mu_i + \delta_i))^{\delta_i})$$

We show these results in Figure 3A and Figure 4B.

The count part of the model is represented by a negative binomial distribution with parameters μ and δ . These do not have a direct interpretation. However, we can compute the expected number of burrows, provided that burrows are present (i.e. conditional that all additional zeros have been removed) from it. Figure 3B and 4C depict the value $E(y_i | y_i > 0)$, which is

$$E(y_i | y_i > 0) = \mu_i / (1 - (\delta_i / (\mu_i + \delta_i))^{\delta_i})$$

Last but not least, Figure 3C and Figure 4D represent the expected number of burrows per plot, considering the probability of occurrence. Values presented in Figure 3C and Figure 4D are calculated by the formula:

$$E(y_i) = (1 - \pi_i) * \mu_i$$

Where

π_i = probability of structural zeros, expected values of zero inflated part of the model
 μ_i = location parameter of the negative binomial part of the model
 δ_i = overdispersion parameter of the negative binomial distribution”

3.) The authors examine 900 plots for the occurrence and number of marmot burrows. They examine each plot twice, once for the period 1968/69 and once for the period 2002-2017. Thus, they have a repeated measures design. It is therefore absolutely necessary to include PlotID as a random effect in their GLMM. So far, data from the same plot in the 1960s and today are treated as independent.

This is an excellent suggestion, thank you! In the revised models, we have included the plot (nested within zone) as a random effect in our model. We revised the Methods section to read: “Because we were specifically interested to quantify the effect of time and land use for both presence and abundance, we included these two variables and their interaction in both the presence-absence and the abundance part of the model. As the plots were surveyed in the historical and the contemporary periods, we corrected for repeated measures by fitting sample plot id as a random effect, nested within study area. Random effects were used to account for unobserved heterogeneity not contained in the covariates..” (Line 261)

Please also see Supplementary Material 6 for the revised model coefficients and Supplementary Material 11 for an explanation of marginal prediction.

4.) In lines 339-345 and Figure S3, the authors present an additional result that is based on additional data on the agricultural history for a subset of 111 plots. This result supports their main finding of a time-delayed marmot decline and is indeed very interesting. However, it remains obscure, what the source for the additional data is, why these additional data are not available for the other plots, and how these additional data were analysed to produce Fig. S3. I suggest that the authors add this information to the Methods section or (at least) provide more details in the supplementary material.

Thank you for your suggestion. In light of this comment and the broader suggestion of the Editor to include further time points in our analyses, we have included a new Methods section describing this (and other) additional analyses. Please see revised Methods section Line 280:

“For a subset of cropland plots, we separated active and abandoned cropland (N=165) and compared burrow density change in relation to transitions between these classes. For these plots we observed no significant difference in the change in burrow numbers between the two classes. Because the separation between fallow and active agriculture is often not possible based on visual image interpretation, and because we observed no significant difference, we combined active cropland and abandoned/fallow cropland into a general cropland class for all subsequent analyses. (Supplementary Material 8).

To identify if the burrow density decline over the 50 years was gradual or abrupt, we carried out two analyses on data subsets. First, for 111 plots where archival map data were available, we compared the average number of burrows in plots where agriculture expanded during the Virgin Lands Campaign (1954-1963) with areas where agriculture was already established prior to the campaign. The cropland extent at the end of the campaign is presented in detail in Supplementary Material 9. For these 111 plots, we compared the average number of burrows in the historical and contemporary time periods amongst two groups: plots that were converted to agriculture during the Virgin Lands Campaign and plots that were cropland already prior to the campaign. Second,

for a subset of 138 plots which were classified as cropland in the contemporary time period, we were obtained additional multi-temporal imagery between 2000 and 2019. We found no significant trend in burrow density between 2000 and 2019, which discounts the possibility of a recent abrupt decline (Supplementary Material 10).”

Furthermore, we have expanded “Supplementary Material 9: Virgin Lands Campaign” to include data description and data coverage of our study area, a new map and a description of the supporting analyses.

5.) The authors use statements such as ‘the declines [in marmot burrow densities] were steepest where cropland use persisted the longest’ (lines 320-321) or ‘the longer cropland persisted, the steeper the declines were in burrow numbers (line 331). These statements are misleading because they imply a gradual relationship between cropland age and burrow loss, where, in fact, there are only two categories (‘persistent cropland’ vs. ‘grassland to cropland’).

We realize that this statement was formulated too generally, while these results really apply only to those 111 plots for which information was available from the time of the Virgin Lands Campaign (see previous comment). To remedy this mistake, we have clarified throughout the manuscript to which of the analyses we refer. Please see our response to the comment above, and more generally the revised results in Line 303 onwards.

6.) In addition to land use type (grassland vs. cropland) or land use change (persistent cropland, persistent grassland, grassland-to-cropland), the authors use a large number of covariates, i.e. additional predictors in the models that shall be controlled, such as soil texture, NDVI, distance to farms etc. These explanatory variables are only very briefly introduced in the main text (267-288); Table S1 does not provide much more information. Furthermore, several covariates have significant effects in the models, but are not discussed at all. If space is really that limited in the main text, at least in the supplementary material the authors should describe each variable in more detail: How was it measured? For which time period? What is the used scale of measurement? In particular it is unclear (a) how soil texture was quantified, (b) whether a contemporary NDVI can be used for the historical period as well, (c) how a ‘river’ is defined here, (d) whether climate change over a half century is indeed negligible, so far this is not convincing.

Thank you for this great suggestion. We have added a new section to the Supplementary Material describing each variable in more detail, included units of measurement to the table and clarified a) the soil texture source and classes used, b) the exact source and values of NDVI, c) the source of the waterway data. For d) we included a reference to the main manuscript indicating that climate change effects in our region are negligible for the study period, especially in relation to the effect of land change. The revised text and table now read:

“Our models included a total of 11 explanatory variables. The two bioclimatic variables (mean temperature of the coldest quarter and precipitation of the driest quarter) extracted from the WorldClim Database [6] for the period 1960-2000 and average for the entire area of the plot. The soil data was extracted from the digitized soil atlas of Kazakhstan, and three classes of soil texture were considered: clay and heavy loam, loam and other [7]. The distance to the nearest river was based on waterway network information extracted from Soviet and Kazakhstan Topographic Maps scaled 1:100,000. The distance to the nearest farm building was also based on

the Soviet and Kazakhstan Topographic Maps scaled 1:100,000 for the historical time period. We verified the contemporary location of the farms in Google Earth imagery, to eliminate from the contemporary variable set the farms that were abandoned following the collapse of the former Soviet Union. . The average slope for each plot, was derived from the Shuttle Radar Topography Mission 90m Digital Elevation Model [8]. The Normalized Vegetation Difference Index (NDVI) values were used as indicators of vegetation productivity, and represented averages during the month of May over the time period between 2008 and 2014. Variables on the period (historical or contemporary), the study zone (central, north or south) and the plot identifier were derived from our own data collection.

Variable name	Variable description	Unit	Source
bio11	Mean Temperature of Coldest Quarter	degrees Celsius	[6], worldclim.org
bio17	Precipitation of Driest Quarter	mm	[6], worldclim.org
d_farm	Distance to nearest farm building	meter	Topographic Maps, Google Earth
d_river	Distance to nearest river	meter	Topographic Maps
dominant_lu	Dominant land use within plot	Cropland/ grassland/ other	Corona, Google, Bing, ESRI
ndvi_may	Normalized Difference Vegetation Index for emergence month of May	NDVI	free.vgt.vito.be
period	Time period of analyses (historical vs. contemporary)	Contemporary/ historical	Corona, Google, Bing, ESRI
slope	Average slope within plot	degrees	srtm.csi.cgiar.org
soil	Soil texture for Kazakhstan	3 classes (see text)	[7]
zone	Area of Kazakhstan in which the plots are located: North, Central, South	N, C, S	
plot	Plot area with diameter of 1km for which burrow number were counted	Unique identifier	

In the revised manuscript, we discussed the effects of other covariates in the Discussion section at Line 359:

“The expansion of the Virgin Lands Campaign in the mid 20th century affected areas with the highest soil quality — areas also preferred by the marmots for the ease of digging. This is a process locally known as ‘colonies absorbed by agriculture’ [53]. Indeed, our modelling results suggested that loamy soils had higher burrow densities compared to clayey or stony soils, which are less favorable for agriculture [48] (Supplementary Material 4). Although bobak marmots

forage preferentially on natural vegetation [26,53], when emerging from hibernation in early spring, natural vegetation is still scarce, and the sprouting wheat may provide an attractive food source for the species. This explanation is supported by our data, which indicates a higher probability of occurrence in areas with high NDVI values specifically in the month of May when wheat typically sprouts (Supplementary Material 4).”

Specific comments by Reviewer 1:

85-92: Not all readers will immediately understand why species populations respond delayed to habitat losses or changes. Could you briefly explain the mechanism? You say that socio-economic pressures might amplify the delay (l. 90-92). Could you give an example?

Thank you for pointing this out. In the revised manuscript, we provided a clearer explanation of the extinction debt / relaxation time concepts, added four new references on this topic and included an example of how historical socio-economic pressures can affect the delay. The revised paragraph now reads:

“Land conversions lead not only to instantaneous, but also time-delayed responses in species richness, distribution, diversity and abundance [17,18]. Delayed species responses to habitat loss have been documented for insects, birds and mammals [19–21]. For mammals and birds, local extinctions or sudden drops in species richness may occur as soon as a few years after land-use conversion, but in plants they can occur also several decades or centuries later [21,22]. Such time-lagged effects may arise from historical and contemporary socio-economic pressures, further causing population declines to lag behind contemporary land conversions. For example, the extinction of some European plant and insect species match historic indicators of socioeconomic pressures more closely than contemporary ones [23,24]. This is why long-term population assessments in relation to land-use histories are essential to fully understand the effects of land conversions on species populations, and to be able to predict future population trends.”

120: 'land-use change'?

Thank you! Changed.

141: 'burrow location' or 'the species' choice of burrow location'?

Thank you! Changed.

148: Here, you mention only Google Earth and Bing, but not ESRI which provided most of the data (line 214).

Apologies for this mistake. Corrected.

line 149, 195, 215 and Fig. 2 (caption and figure): Make time span for contemporary period consistent.

Corrected.

171-173: I do not really understand, why you refer to Fig. 2 here. In Fig. 2, I cannot see which plots or which proportion of plots is located in fallow or abandoned cropland. Is it possible to

distinguish fallow or abandoned cropland from managed cropland or grassland on the satellite images? From the marmots' point of view, is an old field more similar to a cropland habitat or grassland habitat? Would a third category in Fig. 2 and in the analysis be helpful?

Good point. Referencing to Figure 2 at the end of this paragraph was misleading. We moved the reference, and instead provided some further explanation on the distinction between fallow and active agriculture both in the revised Methods section and in Supplementary Material 1. The new text reads as follows:

” Because the separation between fallow and active agriculture is often not possible based on visual image interpretation, and because we observed no significant difference, we combined active cropland and abandoned/fallow cropland into a general cropland class for all subsequent analyses. (Supplementary Material 8)”.

Please see also text at Line 279 and Supplementary Material 8, which contain a further explanation and a new figure on this distinction.

240: Do you mean plots when you say 'samples'? 622 plots out of 900 would be 69.1%, or 622 plots out of 1800 (900 per period) would be 34.6%. How do you get 36%? Ah, you excluded plots with cloud cover, did you not?

Correct. We clarified this in the text by stating the number of plots removed due to cloud cover or missing land use data. The text now reads: “We dropped 37 historical and 43 contemporary plots for which we could not detect land use or burrows due to cloud cover or high image distortion (<5% of all plots).”

244-245: You mean 38 plots were located in fallow or abandoned fields in the contemporary period, but were actively cropped in 1968/69?

Yes, thank you! We rephrased to: ”For a subset of cropland plots, we could separate active and abandoned cropland (N=165; of these 127 plots were actively cropped in both periods and 38 plots were active in the historical period but fallow or abandoned in the contemporary period)”

248: This sounds misleading. Rather say that you want to assess the temporal consistency in the spatial distribution of burrows.

Thank you! Corrected.

253-255: What is the difference between 'probability of philopatry' and 'predicted number of persistent burrows per plot'? Is one term sufficient? I guess you divided the number of paired observations (times two?) by the total number of burrows across time periods? You should make this clear.

We agree this needed more explanation. Please consider also our response to your major comment no 2. Our zero-inflated modelling approach consists of two parts (1- probability of occurrence, 2-number of burrows given that probability is higher than 1). For the philopatry part of the study we assessed 1) how likely it is that persistent burrows occur within one plot over time and 2) how many burrows persisted over time. We tried to make this distinction clear in the text as follows:

“We considered a marmot borrow to represent philopatry if it was found at the same location in both the historical and the contemporary time period. For each plot, we assessed the number and spatial configuration of the burrows over time, to quantify the expected number of burrows lost, gained, and persistent.”

Please also see the explanation of the philopatry model at line 289 onwards.

290-296: I guess, these AIC figures as well as the figures shown in Text S1 refer to your first (and main) model with the number of burrow counts as response. You should make this clear.

Thank you! Corrected.

314-315: This is confusing. Above, you explained that land-use type and time are only predictors in the occurrence and density model, whereas in the models on maintained, lost and newly created burrows, land change is the main predictor. Thus, what do you mean with ‘For all models’ here?

Sorry for not being sufficiently clear here. In the revised manuscript we make a clearer distinction between the main model, which is based on land-use classes and predicts probability of occurrence and predicted density overall, and the three subsequent models that predict occurrence and density of a) lost, b) gained and c) maintained burrows as a function of land-use change (and other variables). The revised text reads:

“To estimate the probability of burrow occurrence and the burrow density per plot, we used a total of 1,720 observations (plots) from both time periods (Supplementary Material 2 and Supplementary Material 5). We estimated the effects of land-use and time on the probability of occurrence and on burrow density, while keeping all other variables at their mean values. (Supplementary Material 6) To estimate the number of burrows that were maintained, lost, and newly created within a plot we used a total of 863 observations paired by plot and time period (Figure 1). For each plot, we considered the major land changes that occurred between the two periods, (persistent cropland, grassland to cropland and persistent grassland), in addition to environmental and anthropogenic co-variates. For each of the three models (maintained, lost, newly created), we estimated the effects of land-change on the probability of occurrence and on burrow density, while keeping all other variables at their mean values (Supplementary Material 7).”

320 + 384 + 428: Actually, from 1968 to 2018, it's 50 years not 60 years.

Thank you! Corrected.

322-324: Leave this for the Discussion.

We agree and removed this part from the Results section and now address this point in the fully revised Discussion section.

334-336: Are these simple observed average values? To which test does the p-value refer? Your occurrence and density model does not differentiate between persistent cropland plots and grassland-to-cropland plots. So where do these numbers come from? Moreover, in the following section (lines 363-365) you give different (but similar) numbers for the same thing... Is this redundancy needed?

Points well taken. In light of your major comments no 2 and 3, we have fully revised the Results section!

402-404: This is not logical. Do you think that burrow densities in your historical grassland plots were untypically low?

Thank you for pointing out this unclarity. The revised manuscript now reads:

“Most notably, the steeper decrease in average burrow densities for plots that were cropped prior to the Virgin Lands Campaign, compared to plots converted to cropland during the Campaign, suggest that marmots display a time delayed response to agricultural conversion, similar to the delayed responses of birds and mammal species [20]. The high densities in historical cropland plots, may also partially still reflect higher densities typical for grasslands, because many of these plots were likely converted from steppe to agriculture during the Virgin Lands Campaign, only a few years prior to our data collection. If marmots respond with a time-delay to agricultural conversion, then burrow densities which were particularly high in historical croplands (compared to historical grasslands), may still reflect densities typical for grasslands prior to conversion.”

Fig. 4A: The labels ‘increase’ and ‘decrease’ need to be exchanged.

Thank you! Corrected.

Suppl. material, line 7: ‘were available’

Revised and removed. Thank you.

Suppl. material, line 65: This should be Table S4

Thank you! Corrected.

Table S1: The ‘www.’ in the source URL for slope and NDVI need to be deleted.

Thank you! Corrected.

Comments by Reviewer 2:

The paper entitled “Cold War spy satellite images reveal delayed declines of a philopatric keystone species in response to cropland expansion” is well written, logical, and interesting. It is novel because the dataset of photos is not yet widely known and used by scientists. Statistical analyses are appropriate, as well as the iconographic part. I have some minor comments below and a general comment about the accessibility of the Corona dataset.

80 add a full stop after “[14-16]”

Thank you! Corrected.

88-89 rewrite this sentence “in mammals and birds” should be placed in earlier position

Thank you! Corrected.

280 add “km” after “17”

Thank you! Corrected.

325 the first sentence of this paragraph is partially a repetition of the paragraph before, please restructure the two paragraphs, or rephrase it to make clear that the first paragraph is a summary of the results

Thank you for pointing this out. In accordance to the Editor’s and Reviewer 1 comments, we have restructured the manuscript considerably, and the revised paragraph now reads:

“Overall, the burrow densities decreased by 14% (N=1,027) since the 1960s (Range: -60 to 55 burrows/plot) and we recorded burrow density decreases in 55% of the plots (Figure 2). Surprisingly, our models revealed that the probability of occurrence was higher in croplands compared to grasslands, independent of time period. After accounting for zero-inflation, over-dispersion, and environmental and human factors that may affect the burrow site selection by the marmots, we predicted higher burrow density in croplands compared to grasslands on average (Figure 3C, Supplementary Material 11).”

Accessibility of the Corona dataset: One of the most important novelty of this manuscript is the use of the database Corona. In the manuscript I did not find a permanent link to the USGS EarthExplorer where the photos can be downloadable. By the way, browsing the Corona dataset in the USGS Earth Explorer is not intuitive and the quality of the photos I have visualized is poor, and certainly below the resolution that they seem to have in Figure 1. This might be due to my not complete knowledge of the way to access the data, but I strongly recommend authors to: 1) clearly indicating a link and brief information on how accessing the aerial photos, and 2) verifying whether the resolution of the photos is reasonable also outside their study area. Due to space limitations, this information could be also in the supplementary materials.

Great suggestion, thank you! We have included the link to the data in the manuscript at line 189 as well as new section in the Supplementary Material 1, which details the accessibility and availability of the Corona data. We also include there further references to papers which detail recent developments in image-processing of Corona from other parts of the world and a list of images that were used in our study, with step-to-step instructions on how to download them from the USGS EarthExplorer website.

“Supplementary Material 1: Corona imagery

Corona spy satellite imagery represents one of several spy satellite data collections from the Cold War period, when the US government initiated multiple space-borne photography missions for intelligence purposes [1]. The data have been gradually declassified since 1996, and are now available via <https://earthexplorer.usgs.gov> (under Declassified Data tab). Data coverage is global [2], but most imagery were collected in areas of the former Soviet Union. Due to its experimental nature, the data vary in extent, temporal and spatial resolution, but high-resolution imagery (2-10m) has successfully been identified and used in multiple parts of the world already. All data previously acquired are freely available for download on the USGS website. [3–5]

In this study, we only use stereo-high and stereo-medium Corona imagery (see Additional Criteria Tab on <https://earthexplorer.usgs.gov>) with no or low cloud coverage. We assessed the spatial extent of the data and the cloud cover based on freely available thumbnail images and purchased 12 pairs of stereographic images dated September 1968 and 1969, with an average ground resolution of 2.3m, all listed in the table below. To download the raw Corona data, please use the <https://earthexplorer.usgs.gov/> website, selecting Declass 1 (1996) in the Data Sets tab and entering the Image unique identifier from the table below into the Entity ID column of the Additional Criteria Tab. “

Please also see the new table in Supplementary Material 1.

Appendix B

Response to Editor and Reviewers

Comments by Editor:

Dear authors,

The most critical previous reviewer and myself have read the revision in detail. We think it has improved in clarity, but still think that quite some more work is needed. Yes, the topic is very suitable for the journal, we are all enthusiastic about the design of the study, very nice long-term dataset and novel use of spy satellite data, but the interpretation of results and Discussion is still not the most insightful and lacks critical dissemination.

Specifically, the reviewer highlights that the discussion provides very little insights into the mechanism underlying the differential temporal responses in grass and cropland. In addition, I also am totally not convinced about the strong emphasis in the MS for a time-delay in the response and how this is inferred from the analyses. The reviewer and myself also have a 1-2 other major, but otherwise mostly minor comments that also need to be addressed, and the text needs to be thoroughly checked for language (we have only highlighted some of the errors we encountered).

Overall, I am quite ambivalent about what to do with this MS. Both reviewers liked the MS very much and emphasized that it is important to be published, but at the same time the revision struggles to get the Discussion up to standard for our journal in my opinion. The work that needs to be done clearly falls in the major revision category as it touches on the key message of the MS, even if it may mainly concern rewriting of some parts of the Discussion. Technically, journal policy actually does not allow for a second round of revision. Notwithstanding, I recommend to the senior editor to seriously consider allowing for another round of major revision in this specific case, but I can see it go either way.

Thank you again for the very helpful and constructive reviews of our paper, as well as for the opportunity to re-revise the manuscript. The comments of the reviewers and editor were very helpful, and we feel that our manuscript has again improved substantially.

The major changes to our manuscript are summarized here, but please see the detailed revisions in the response section below, as well as in the manuscript.

- We fully revised our interpretation of results and the Discussion section to de-emphasize the time-delayed response. In the revised manuscript, we now focus rather on the gradual decline in response to agricultural conversion and only indicate that this may lead to time-delayed responses, which need to be further investigated.
- We provided a more comprehensive discussion of the possible mechanisms underlying the slow, gradual decline we observe. We explain how marmots may respond to cropland conversion, intensification, pesticide, hunting or disease and discuss the evidence towards these lines of explanation.

- We revised the statistical analyses, to include the proportion of disturbed burrows rather than absolute numbers, as suggested by reviewer 1. The main conclusions remain unchanged, but the predictive power of the models has improved substantially. Thank you for this suggestion!
- We highlighted the contribution of our manuscript over a previous study that only assessed marmot population at one, recent point in time, and expanded our discussion of our results in light of this study.
- We have included a caveats section to the Discussion.
- We thoroughly checked for language and had a native speaker proofread the manuscript.

Major comment AE:

1. The discussion starts by stating “We reveal one of the longest recorded time-delayed response of a mammal to agricultural conversion (up to 50 years)” The “time-delay” in the response is not mentioned in the Results at all, so it comes out of the blue here as one of the main conclusions of the MS. The results show that declines were most prominent in persistent croplands and in those plots where cropland use persisted the longest. The discussion should then first explain why this can be interpreted as a delayed response and what the caveats are. By interpreting the stronger decline in persistent croplands compared to more recent croplands to a time-delayed response, the authors interpret this as a time delayed effect. But croplands converted earlier and later may differ in many ways (the authors suggest that they converted the best lands first), and it is unlikely that all of the factors may have been accounted for in the models (the agricultural regimes were not experimentally randomized treatments, which limits the inference that can be drawn, which should be acknowledged in the MS).

Thank you for this valuable comment. In retrospect, we agree that we overly emphasized the time-delayed responses. We agree that the time-delayed response is only one potential explanation for the patterns we observe, and more investigations are needed to support it. Consequently, we have rewritten our Discussion section entirely to address the concerns of the Editor and Reviewer 1. In the revised manuscript, we start the Discussion by highlighting that the decline observed in burrow numbers was slow and gradual over the 50 years - rather than abrupt – and continue to provide explanations on the mechanisms behind these declines.

Line 348: “Impact assessments of agricultural expansion on biodiversity typically focus on the time immediately following habitat loss, which is problematic if biodiversity changes are gradual over long time periods. We reveal one of the longest recorded responses of a mammal to historical agricultural conversion and highlight that single snapshots in time may provide insufficient information for understanding how species respond to land conversions. Our analysis of changes in marmot burrow densities since the 1960s suggests that bobak marmot populations declined as a result of past habitat conversion, and that these declines occurred on timescales of up to 50 years. Burrow declines were steepest in persistent cropland, indicating that declines are a long-term, gradual response to historical agricultural conversions [38], related to repeated and increased burrow disturbance and reduced food availability [23,49].”

[...] please see full discussion of possible mechanisms causing this decline (response below), as well as alternative explanations in the revised manuscript

Line 367: “It is likely that extensive agricultural practices - common in Kazakhstan until the early 2000s [10,40] - reduced forage availability during the fattening season, which in turn prevented marmots from gaining sufficient body mass to survive hibernation [23,52]. Taken together, the persistent cropping over 50 years, coupled with high rates of burrow disturbance and reduced forage may explain the observed long-term, gradual population decline. Although historical agricultural regimes could not be experimentally randomized across our study area, the Virgin Lands Campaign represented possibly the largest natural experiment on the effects of agricultural conversion for biodiversity, and our results support the idea that an increase in the frequency of system disturbance can lead to long-term population declines [22].”

Also how does the delay work mechanistically is not really explained. In the intro it is suggested that ‘Land conversions lead not only to instantaneous, but also time-delayed responses in species richness, distribution, diversity and abundance.’ But there may also be difference in changes over time in intensity of land use between these type of croplands (I am not an expert on farming in Kazakhstan, but increased plowing, pesticides, earlier converted may have depleted the resources earlier thereby increasing needs for fertilizer). I do not necessarily doubt the interpretation of the authors, just that the delayed response is presented as a given with very little critical discussion and suggestion for mechanisms.

Thank you for these thoughts. In the revised discussion we address the potential mechanisms behind instantaneous vs. gradual or time-delayed responses to agricultural practices (see Line 348, in above response) and Line 358:

“We showed that declines in burrow densities were steeper in persistent cropland compared to persistent grassland, and in plots that were cropped prior to the Virgin Lands Campaign, compared to plots converted later. The repeated disturbance of burrows through plowing, likely led to increased colony stress and higher energy costs for re-establishing disturbed burrows [23,50,51], ultimately reducing colony fitness and size [23]. Because the declines were steepest in older fields, repeated disturbances associated with cropping may substantially decrease population size over time, despite the effects of single disturbances possibly being minor [22]. Additionally, agricultural conversion likely reduced the forage quantity and quality for the marmots, which preferentially forage on natural vegetation [26,51]. It is likely that extensive agricultural practices - common in Kazakhstan until the early 2000s [10,40] - reduced forage availability during the fattening season, which in turn prevented marmots from gaining sufficient body mass to survive hibernation [23,52]. Taken together, the persistent cropping over 50 years, coupled with high rates of burrow disturbance and reduced forage may explain the observed long-term, gradual population decline.”

Furthermore, we clarify that intensification is only a recent process in Kazakhstan (Line 388) and that to this point no immediate effect of intensification could be identified.

“An alternative explanation to the gradual, long-term decline we observe is that land-use intensification led to drops in marmot population due to indirect effects of pesticides and herbicides [24]. This drop however, would have been recent and more abrupt, because

intensification in Kazakhstan only started in the early 2000s, when over 2 million ha of cropland transitioned to no-till, and imports of herbicides increased substantially [10,40]. Pesticides and herbicides can affect marmots through direct contamination and by reducing forage availability during the fattening season [53]. Although preliminary field-data suggested that marmot colonies have disappeared in some croplands where no-till (and thus heavy pesticide use) has been adopted, the average numbers of burrows per plot since 2000 did not change significantly (Supplementary Material 10), rendering intensification an unlikely explanation for the strong declines we found. However, systematic assessments of herbicide impacts over longer time periods would be beneficial to elucidate if and on which time scales pesticides may affect population dynamics of burrowing mammals. “

L383 here you write ‘if marmots respond with a time-delay to agricultural conversion’ It seems the authors themselves also think here that the conclusion of a time delay is not strongly supported yet. In addition, testing for a time delay was also not mentioned as a study aim at the end of the Introduction, which suggest to me the study did not set out to look into this. Why is it then so prominently emphasized (e.g. in title of MS)?

We agree and have removed the time-delayed aspect from the title, and now only mention it as one possible explanation for the observed decline in burrow numbers.

Please see Line 417 ” High burrow densities in historical croplands, shortly after the end of the Virgin Lands Campaign, suggest that burrow numbers did not drop immediately following conversion, further offering evidence for a gradual, possibly time-delayed response to agricultural expansion [18,53]”

2. L208. I was quite surprised to read here about this other paper [ref 48] by some of the same authors on using satellite imagery on the same species and location on a very similar questions. Why is this paper not mentioned in the Introduction (and cover letter for that matter), and explained how the current MS advances or is different from the previous work?

We apologize for the misunderstanding regarding reference [48]. This reference was outdated in our initial manuscript, but identical to reference [34] of the first submission (and reference 33 of the revised manuscript), which was cited both in the Introduction and Methods. We do apologize for not discussing the reference 34, sufficiently in the context of our manuscript. We have corrected this.

We would like to highlight that the main contribution of Koshkina et al was to test the feasibility of using burrow numbers extracted from satellite imagery to infer population estimates. While the Koshkina study made this fundamental contribution, the work lacked any temporal perspective.

In the revised manuscript we cite Koshkina et al. is at Line 111 as an example of using remote sensing data to map marmot burrows. We also highlight here the remote sensing data limitation.

“However, satellite imagery with resolutions high enough to detect burrowing animals is typically only available since the 2000s, precluding long-term studies”

Also, the MS now suggests there were no false positives in the validation study, but ref 48 states the opposite.

Thank you for pointing out this omission on our end. We revised the text to clarify that no false negatives occurred and that false positive were only common in areas with abandoned colonies.

“Burrow location validation with field visits suggested that no false negatives occurred [33]. False positives only occurred in recently abandoned colonies (where burrows are usually covered by darker vegetation than the surrounding areas), but these were extremely scarce in our study area [33]. Overall, only ca. 40% of the burrows on the ground are detectable with remote sensing, likely because temporary summer burrows are small [33]. In total, 36% of our samples (622 plots) had burrows.”

Most importantly, ref 48 suggest the detection bias depends on land use (their fig. 3 below). There seems to be a very large difference in underestimation from satellite data for example between grazed (GR) and ungrazed (UN) grasslands. The current study does not distinguish grazed and ungrazed grassland and bins them and studies burrow densities using satellite data over time. At the same time the authors state that grazing has increased over time. So can land use -dependent detection and land use change within grassland and cropland affect the results on changes in burrow densities over time? Finally, in the validation a critical (but untested) assumption is made that the detection bias patterns from is the same in recent and cold war satellite data. Clearly more critical discussion is needed. Also L302: Do these numbers account for imperfect detection and differences in detection probability among habitat? L333 How is this analyses affected by incomplete detection?

Thank you for this valuable comment. Due to image quality and lack of field-validation data for Corona imagery, we unfortunately could not reliably differentiate the land-use classes of grazed and ungrazed steppe, and therefore merged them into one single grassland class. We believe that this might have led to our estimates of densities in steppe/ grasslands to be conservative. We included this caveat in our Discussion section:

Line 432: “Finally, we caution that our study could not differentiate between grazed and ungrazed steppes, both combined in our single “grassland” class. Analyses of contemporary imagery suggest that burrow detection probability for ungrazed steppes (19%) is lower than for grazed steppes (46%) [33], which means that our estimates of burrow densities in grasslands may be conservative. Although our analyses could not account for detection bias statistically, because ground validation data was not available for the historical time period, we would expect detection rates between land uses to be similar across time periods. It is possible however that estimates for the historical time period are generally conservative, because overall image quality is lower compared to recent imagery. This suggests that the estimated magnitude of the decline is also conservative.”

Minor comments:

L132 relocation. Rerword? You do not follow individual marmots/burrows over time, so if a burrow disappeared you do not now whether it relocated or went extinct.

Rerworded to spatial redistribution of burrow locations.

L155 his->the

Changed.

L160 add comma after Union

Changed

L180 To me the usage of 'acquired' suggest thre photograph were bought in 1968, do you mean 'taken'

Reworded

L296 delete 'were''

Deleted.

L307 what model are the authors referring to here? L310 'predicted', do you mean the estimated difference between habitats from the data while accounting for other confounding effects in the statistical model? Predicted suggest to me this was an expectation, but it is in fact a result if I understand it correctly.

Apologies for this misunderstanding. We refer to model predictions, of course. We changed the terminology to “estimated” to avoid confusion.

L327 inconsistent use of abbreviation VLC

Corrected.

L329 marmots-> you do not analyze marmots, but marmot burrows. Equating burrow philopatry with marmot philopatry is more something for the Discussion I suggest.

Thank you for this valuable comment. We have checked our use of the term philopatry throughout to highlight that the persistence of a burrow in exactly the same location is a sign of colony-level philopatry (not necessarily individual-level philopatry). Furthermore, we highlighted that burrow persistence does not indicate individuals, but these measures are expected to be highly correlated.

Line 213:” We considered a marmot burrow to indicate philopatry if it was found at the same location in both the historical and the contemporary time period.”

Line 326: “Our analyses revealed remarkable long-term persistence of marmot burrows despite drastic land-use change, suggesting a high degree of site-conservatism and philopatry in steppe marmots.”

Line 407: “We caution that our study quantifies the persistence of burrows, not the philopatry of individuals themselves, but we expect these measures to be strongly correlated.”

L347 burrow philopatry

Corrected

Comments by Reviewer 1: *The authors have substantially revised their manuscript, including the statistical analyses. They have also added some extra analyses. The revised and new results strengthen the previous conclusions of the authors. Most important, the decline in marmot burrows is now restricted to cropland, while in grassland, the expected number of burrows even increased. The revised Figs. 3 and 4 are better to interpret now, in particular together with explanations provided in the new Supplementary Material 11, which is very helpful. The provided explanations for the counterintuitive higher burrow densities in croplands compared to grasslands are reasonable. Overall, the manuscript has gained much in clarity.*

Although the authors' line of thought is more convincing now, I still have some doubts concerning the assumed delayed response of the marmot population to agricultural expansion (see first and third comment below). Moreover, the clarity of the revised manuscript has opened my eyes for another shortcoming of the statistical analysis (comment 2). However, I would like to stress that, in my opinion, a re-revised version of this manuscript would make a great contribution to RSPB.

Thank you for your thorough and thoughtful comments, which once more helped us to improve our manuscript. We address each of your suggestions below.

1. The authors assume that the decline in marmot burrows on cropland over the last 50 years is a delayed response to the conversion of steppe grassland to cropland during the Virgin Lands Campaign (1954-1963). However, the mechanism by which agricultural land-use caused declines in marmot populations remains obscure in the manuscript for a long time. This is not at all clear given that cropland features a much higher burrow density than grassland (Fig. 3). In contrast, the authors stress the good food resources on cropland (l. 364-371, 416-417). Only in the second half of the Discussion, the authors explain, why they assume detrimental effects of land conversion on marmot populations (unfortunately without citing any reference): 'The repeated disturbance of the burrows through agricultural practices (i.e. tillage, harvest, pesticide application) might lead to population fitness declines, ultimately causing a population drop (418-420).' I suggest to move this part to the Introduction, to elaborate a little bit more on this and include other evidence for negative effects of agricultural practices on marmots. This would make the reader less doubtful on the authors' assumption.

Thank you for your valuable suggestions. In fully revising our discussion section, we have addressed your concerns in several ways. In addition to the points below, please also see our response to the Major Comments of the Editor on mechanisms and delayed responses.

In the revised discussion we highlight that our results indicate a “gradual, long-term decline in the population” (rather than a time delayed response) and discuss possible mechanisms behind this gradual decline (Line 350 onwards).

We discuss the mechanisms behind a potential gradual, slow decline at Line 358 onwards (see response to Editor comments) and discuss possible effects of hunting, disease and pesticide use at Line 371:

“In addition to cropland conversion, disease, poisoning, hunting and trapping could have contributed to the gradual, long-term population decline we observed. Although it is possible that the effects of cropland conversion were locally modulated by these factors, effects of disease and poisoning are unlikely to be substantial at the spatial and temporal scale of our study. Parasites and disease may cause local mortality in marmot populations, but no major demographic effects have been reported for the bobak marmot in Central Asia since the mid-20th century [23]. Poisoning of burrowing mammals has been a common practice historically in parts of Canada, US and Mexico, but was not widely practiced in Kazakhstan [23,28]. Bobak marmot populations have been historically affected by overhunting and trapping, especially in Russia, but since the 1950s hunting became regulated and the marmot population rebounded [26,50]. Furthermore, fur trapping and hunting are highest in proximity of human settlements, so their effects would be partially accounted for in our analyses via the predictor distance to farm [50].

An alternative explanation to the gradual, long-term decline we observe is that land-use intensification led to drops in marmot population due to indirect effects of pesticides and herbicides [24]. This drop however, would have been recent and more abrupt, because intensification in Kazakhstan only started in the early 2000s, when over 2 million ha of cropland transitioned to no-till, and imports of herbicides increased substantially [10,40]. Pesticides and herbicides can affect marmots through direct contamination and by reducing forage availability during the fattening season [53]. Although preliminary field-data suggested that marmot colonies have disappeared in some croplands where no-till (and thus heavy pesticide use) has been adopted, the average numbers of burrows per plot since 2000 did not change significantly (Supplementary Material 10), rendering intensification an unlikely explanation for the strong declines we found. However, systematic assessments of herbicide impacts over longer time periods would be beneficial to elucidate if and on which time scales pesticides may affect population dynamics of burrowing mammals.”

Further, following your suggestion, we mention mechanisms that may cause declines in population in the Introduction, to establish context:

Line 84 “Land conversions can lead to gradual or time-delayed declines in richness diversity and abundance because species may require some time following disturbances, until they reach a new equilibrium [15,16]. Land conversion can create population sinks, where local extinctions occur within years [17–19], decades, or centuries [19,20]. The speed and timing of population declines may depend on the spatial configuration of remaining habitat and life-history traits, such as longevity [21]. Furthermore, agricultural practices may affect population fitness and lower forage availability leading to lower recruitment or survival over time [22–24]. This is why long-term population assessments following historical land conversions are essential to understand their full effects on species.”

Line 96: “Rodents are a food source for larger predators, and through digging and herbivory, they increase soil nitrogen content and forage quality for large grazers [25]. However, human activities have caused major declines in burrowing rodent populations worldwide, directly through poisoning or hunting, and indirectly through agricultural expansion and intensification [23,28]. The repeated disturbance of the burrows through agricultural practices (i.e. tillage, harvest, pesticide application) might lead to population fitness declines, ultimately causing a

population drop [22,23]. Many burrowing rodents exhibit philopatric behavior [29], meaning that their dispersal is constrained either by life history or ecological factors..”

2. The authors observed a remarkable philopatric behaviour in bobak marmots. Moreover they found ‘a higher proportion of burrows retaining their exact location in undisturbed grassland habitat than in croplands’ (l. 348-349). This is used as an argument to support their assumption that agricultural practices lead to the long-term population decline (l. 409-422). Although this conclusion is probably correct, the authors did not formally test, whether the proportion of persistent burrows is higher in grassland than in cropland. What they tested is whether the absolute number of persistent burrows is higher in grassland than in cropland, which is not the case (Fig. 4). The same is true for the number of lost and gained burrows. Since the number of burrows is generally higher in cropland than in grassland, these tests are misleading. I suggest, that the authors use the proportion of persistent, lost and gained burrows as response variable in their models, instead of the absolute numbers.

Thank you for this valuable suggestion. To address this comment, we have revised our statistical models of lost, persistent and new burrows to account for the number of burrows that a plot started out with in the historical time period. While the general patterns stayed the same, and the revised results still indicate higher rates of philopatry in persistent grasslands compared to persistent croplands and the model performance has improved considerably (see table below)

Model	AIC revised model	AIC initial model
Lost burrows	1996.90	2789.00
Persistent burrows	1967.60	2442.70
New burrows	2439.00	2694.50

The manuscript now includes revisions

- in the **Methods** section at:

line 217: “We considered a marmot burrow to indicate philopatry if it was found at the same location in both the historical and the contemporary time period. For each plot, we assessed the number of burrows lost, persistent and new in relation to the number and location of burrows in the historical time period.”

- In the **Results** section at:

line 336: “Our models predicted that persistent cropland plots (i.e. plots that were converted to cropland during or prior to the Virgin Lands Campaign) lost a higher proportion of burrows (62% +/- 6%) compared to stable grasslands plots (40% +/- 5%), and had a lower proportion of maintained burrows (Figure 4D). Specifically, for a hypothetical plot that had 19 burrows initially, we estimated that in persistent grasslands approximately 33% of the historical burrows were maintained, suggesting philopatry of their denizens compared to only 22% in croplands (Figure 4D, Supplementary Material 7 and Supplementary Material 11). This relationship was consistent, regardless of the initial burrow number, but the differences were even bigger for plots which had higher numbers of initial burrows (Supplementary Material 7). “

- In the **Discussion** section at:

Line 402: “Despite the overall reduction in marmot burrows observed here, many individual marmot burrows persisted for approximately 50 years. This persistence is remarkable for a species with life expectancy ranging between 5-7 years [23]. A higher proportion of burrows

retained their exact location in undisturbed grassland habitat compared to persistent croplands. Without disturbance, marmots tend to reuse the same wintering burrows for multiple years and spend less than four minutes per day maintaining their burrow [23], sometimes only changing the main entrance and the mound [51] – suggesting that our estimate of philopatry is likely conservative. The substantial past investment in burrow systems [23], combined with attractive early spring food availability from sprouting wheat [38] and potential competition for remaining suitable habitat may compel marmots to remain in suboptimal cropland habitat. However, because rates of burrow persistence were lower in cropland plots than in grassland plots, we suggest that philopatry in conjunction with the long-term agricultural use might create an ecological trap for the species in cropland fields [30]. We caution that our study quantifies the persistence of burrows, not the philopatry of individuals themselves, but we expect these measures to be strongly correlated.”

Last but not least, we have updated **Figure 4** as well as the figure caption, and fully revised **Supplementary Material 7** to include new model coefficients, and further explanations on how the number of initial burrows affects the philopatry across a range of values.

3. The authors mention that ‘low-intensity agricultural practices’ were ‘historically common in Kazakhstan’ (l. 369-370). Does this mean that agriculture has been intensified during the last 50 years (at least on the most suitable sites where it was not abandoned)? Could this be the reason for the marmot decline?

Thank you for this comment. In retrospect we realize that the issue of intensification was not clearly addressed in our manuscript. Indeed, intensification of agriculture only began in Kazakhstan in the early 2000s and we have found no immediate response to those practices in our ancillary analyses which looked at burrow change between 2000-2019 (Methods line 285). To clarify this, we made following revisions to the manuscript:

Introduction at Line 123: ‘Finally, substantial recultivation of abandoned fields and cropland intensification occurred after 2005 [10,40], due to policy reforms and rising global cereal prices [38].’

Study are at line 164: “Many areas have been re-cultivated in recent decades following increasing world market prices for cereals, improved institutional conditions and technological progress, such as the adoption of no-till agriculture [10,40]. The Post-Soviet abandonment trend may provide new opportunities for steppe conservation, but these are diminishing as agricultural re-cultivation and transition to no-till agriculture have increased since the early 2000s [38,41].”

Most importantly, we clarify this in the Discussion, Line 388:

“An alternative explanation to the gradual, long-term decline we observe is that land-use intensification led to drops in marmot population due to indirect effects of pesticides and herbicides [24]. This drop however, would have been recent and more abrupt, because intensification in Kazakhstan only started in the early 2000s, when over 2 million ha of cropland transitioned to no-till, and imports of herbicides increased substantially [10,40]. Pesticides and herbicides can affect marmots through direct contamination and by reducing forage availability during the fattening season [53]. Although preliminary field-data suggested that marmot colonies have disappeared in some croplands where no-till (and thus heavy pesticide use) has been adopted, the average numbers of burrows per plot since 2000 did not change significantly

(Supplementary Material 10), rendering intensification an unlikely explanation for the strong declines we found. However, systematic assessments of herbicide impacts over longer time periods would be beneficial to elucidate if and on which time scales pesticides may affect population dynamics of burrowing mammals.”

4. The language of added or revised text sections (main text and supplementary material) does not meet the standard of the journal and needs editing.

We apologize for this shortcoming. We have corrected our language throughout and had a native English speaker thoroughly check our grammar and spelling.

Specific comments

1-2: The original title was shorter and better.

Response: Retained the original title, but removed the time-delay, as suggested by the Editor.

84-87: Socio-economic pressures may result in land conversions and associated habitat loss or habitat degradation, which in turn lead to local species extinctions. But why do species respond delayed and not immediately? This cannot be explained with socio-economic pressures but only with the species' ecology, e.g. traits like longevity, philopatric behaviour, slow population dynamics.... Please, provide one example, at best for a mammal.

We revised the text to read: Land conversions can lead to gradual or time-delayed declines in richness, distribution, diversity and abundance because species may require some time after a disturbance until they reach a new equilibrium [15,16]. Land conversions can turn habitat areas in population sinks, where extinctions occur as soon as several years after conversion [17–19] up to decades or centuries later [19,20]. The speed and timing of population declines may depend on the spatial configuration of remaining habitat and life-history traits, such as longevity [21]. Furthermore, agricultural practices may affect population fitness and the lower forage availability can lead to lower recruitment or survival [22–24]. This is why long-term population assessments in relation to land-use histories are essential to understand the effects of land conversions on species populations.

Please note that this section includes new references.

131-132: This final expectation is not comprehensible. Better leave as in the original manuscript.

Thank you. We retained the original text.

134-135: “marmot burrows, and related them to the surrounding”

Changed.

155-156: “this campaign”

Changed.

271-278: The terminology and the order of terms (maintained, lost, newly created) differ from those in l. 229-232 (lost, gained, persistent). Be consistent throughout the manuscript and the supplementary material.

Apologies for this inconsistency. Corrected.

302-305: Omit these general statements without any reference. They are redundant anyway.

Removed in the process of fully revising the discussion.

305: It would be interesting to get to know the average burrow densities in both time periods. The percent decrease can then be set in parentheses. Moreover, given the large range in decreases or increases from -60 to +55 burrows/plot, an average decrease be 14% might not be significant. I guess you have tested this decrease independent of land-use with a more simple GLMM?

We apologize for this misunderstanding. The 14% decline (Range: -60, +55) represents a summary of the raw data. The modelled figures are presented in Figure 3. We clarified this:

(14% of the observed historic number of burrows)

Overall, our dataset indicated that burrow numbers decreased by 14% (N=1,027) since the 1960s (Range: -60 to 55 burrows/plot) and we recorded burrow density decreases in 55% of the plots (Figure 2).”

We included the average burrow densities at line 211: “Plots where burrows were present had an average density of 18.2 burrows/plot (19.4 for the historical periods, 17.1 for the contemporary period).”

312: Where does this 43% come from? According to Fig. 3B the decline is about 30% and according to Fig. 3C the decline is about 60% (as you write in line 317).

Corrected to read: “However, most of the decline occurred in croplands, where the expected number of burrows dropped from 8.43 (+/- 2.3) burrows compared to 3.35 (+/-1.2) burrows since the historical time period (Figure 3C), even after accounting for zero-inflation and overdispersion. Our model predicted a very small gain in grasslands (on average, less than 1 additional burrow per plot) (Figure 3C, Supplementary Material 6 and Supplementary Material 11). This suggests that approximately 60% of the historical burrows were lost in croplands, whereas grasslands that persisted since the 1960s gained about 17% of the historical burrows.”

322-323: To which ‘pattern’ do you refer here? So far you have not described any pattern that could be confirmed. What you describe in Supplementary Material 9 is the pattern itself.

Rephrased to: “Using ancillary information on cropland use prior to the Virgin Lands Campaign, we estimated that 17% of the agricultural fields identified in Corona images were used for crops prior to the Virgin Lands Campaign”

329-330: The ‘massive agricultural expansion’ occurred before the historical time period and not between time periods. Do not confuse the reader.

Rephrased to: Albeit land use change, the majority of plots we assessed had at least some burrows at exactly the same locations as in the historic period.

Suppl. Mat. 9: You analysed 36 preVLC and 95 VLC plots (together 131), but talk about a total of 111 plots for this analysis?

Apologies for this typo. We have analyzed 111 plots (95 VLC and 16 preVLC) – we have corrected this mistake in the revised version, including at Line 334 or revised manuscript.

Fig. 3 + 4: Refer in the caption to Suppl. Mat. 11

Due to space limitations, we did not refer to the Supplementary Material in the figure caption, however, we thoroughly reference it throughout the text whenever we reference the figure themselves: e.g. Line 335, Line 339, Line 369.

Suppl. Mat. 9: Have you tested the decline in mean burrow numbers statistically? How?

We have included information on the statistical test to Supplementary material 9. We relied on the Tukeys test of means to compare the average number of burrows across time periods.